# A dynamin 1-, dynamin 3- and clathrin-independent pathway of synaptic vesicle recycling mediated by bulk endocytosis

**Yumei Wu[1,2,5], Eileen T O'Toole[3], Martine Girard[4], Brigitte Ritter[4†], Mirko Messa[1,2,5], Xinran Liu[1], Peter S McPherson[4], Shawn M Ferguson[1,2], Pietro De Camilli[1,2,5]\***

[1]Department of Cell Biology, Yale University School of Medicine, New Haven, United States; [2]Program in Cellular Neuroscience, Neurodegeneration and Repair, Yale University School of Medicine, New Haven, United States; [3]Department of MCD Biology, University of Colorado, Boulder, United States; [4]Department of Neurology and Neurosurgery, Montreal Neurological Institute, McGill University, Montreal, Canada; [5]Howard Hughes Medical Institute, Yale University School of Medicine, New Haven, United States

**Abstract** The exocytosis of synaptic vesicles (SVs) elicited by potent stimulation is rapidly compensated by bulk endocytosis of SV membranes leading to large endocytic vacuoles ('bulk' endosomes). Subsequently, these vacuoles disappear in parallel with the reappearance of new SVs. We have used synapses of dynamin 1 and 3 double knock-out neurons, where clathrin-mediated endocytosis (CME) is dramatically impaired, to gain insight into the poorly understood mechanisms underlying this process. Massive formation of bulk endosomes was not defective, but rather enhanced, in the absence of dynamin 1 and 3. The subsequent conversion of bulk endosomes into SVs was not accompanied by the accumulation of clathrin coated buds on their surface and this process proceeded even after further clathrin knock-down, suggesting its independence of clathrin. These findings support the existence of a pathway for SV reformation that bypasses the requirement for clathrin and dynamin 1/3 and that operates during intense synaptic activity.

**\*For correspondence:** pietro.decamilli@yale.edu

**Present address:** †Department of Biochemistry, Boston University School of Medicine, Boston, United States

**Competing interests:** The authors declare that no competing interests exist.

## Introduction

Synaptic vesicles (SVs) are the specialized organelles that store and secrete non-peptide neurotransmitters at synapses. Following exocytosis, their membranes are rapidly recaptured by endocytosis and recycled for the generation of new fusion-competent SVs (*Ceccarelli et al., 1973*; *Heuser and Reese, 1973*; *Betz and Bewick, 1992*). Strong evidence indicates that clathrin-mediated endocytosis (CME) plays a key role in this endocytic traffic (*Heuser and Reese, 1973*; *Maycox et al., 1992*; *Granseth et al., 2006*; *Heerssen et al., 2008*; *Kasprowicz et al., 2008*; *Saheki and De Camilli, 2012*). Endocytic vesicles generated by CME have a small homogenous size, in the same range of SVs (*Shupliakov et al., 1997*; *Raimondi et al., 2011*). While it had been originally hypothesized that such vesicles transit through an endosomal station (*Heuser and Reese, 1973*), subsequent studies have suggested that they may mature directly into SVs, a possibility consistent with their size (*Takei et al., 1996*). However, CME does not account for all SV recycling and the mechanisms of alternative pathways have been the object of intense debate.

One proposed alternative recycling mechanism is so-called 'kiss-and-run', a process whereby SVs fuse only very transiently with the plasma membrane without collapsing into it, so that the SVs reform directly by the immediate closure of the fusion pore (*Fesce et al., 1994*; *Alabi and Tsien, 2013*). Yet another mechanism is bulk endocytosis, a high capacity pathway of plasma membrane internalization

**eLife digest** Neurons propagate electrical signals from one cell to the next using small molecules called neurotransmitters. These molecules are held inside small compartments called synaptic vesicles. Once a neuron receives an electrical stimulus, the membranes that enclose the synaptic vesicles fuse with the plasma membrane that encloses the neuron. This releases the neurotransmitters, which then trigger an electrical signal in the neighboring cell. Once the neurotransmitters are released, the vesicle membrane is rapidly reinternalized from the plasma membrane in a process called endocytosis and then recycled, ready for the next round of signal transmission.

The process of synaptic vesicle membrane endocytosis and recycling has been studied extensively, and several different mechanisms by which it occurs have been identified. The best understood relies on a protein called clathrin, and is thought to be essential for nervous system function. Recently, however, a mechanism of vesicle membrane endocytosis that does not involve clathrin was identified. This mechanism, called bulk endocytosis, involves reinternalizing large regions of the cell plasma membrane to generate large compartments called vacuoles, from which new synaptic vesicles eventually form. This mechanism has been observed when neurons fire at high frequency. The cellular processes underlying bulk endocytosis are not well understood, although several studies suggest proteins called dynamins are important.

Wu et al. simulated the conditions a cell experiences during high levels of activity in neurons that lacked the two major dynamins present at the synapses between neurons—dynamin 1 and dynamin 3. In these neurons, robust bulk endocytosis occurred, suggesting that these two major neuronal dynamins do not play a role in this process. Furthermore, formation of vesicles from the vacuoles generated by bulk endocytosis appeared to be clathrin-independent. These findings point to the occurrence of a pathway of synaptic vesicle recycling that bypasses the need for dynamin 1 and 3 as well as for clathrin.

To reconcile these results with previously published work, Wu et al. propose that dynamins may only be required for processes that also require clathrin. But how are vesicles recycled during bulk endocytosis if dynamins are not involved? There are currently few leads to base alternative mechanisms on. Further work is required to unravel this mystery, and to provide insights into how clathrin-dependent and independent recycling processes are linked during high neuronal activity.

involving patches of membrane larger than the membrane of individual SVs. This mechanism predominates under intense neuronal activity (*Miller and Heuser, 1984*; *Holt et al., 2003*; *Wu and Wu, 2007*; *Hayashi et al., 2008*; *Cousin, 2009*; *Wenzel et al., 2012*; *Kittelmann et al., 2013*). Endocytic intermediates generated by bulk endocytosis, henceforth defined as 'bulk' endosomes, eventually disappear as new SVs appear, suggesting a precursor-product relationship and possibly a direct formation of SVs from bulk endosomes (*Heuser and Reese, 1973*).

Studies in *Caenorhabditis elegans* have opened new questions on SV recycling mechanisms. Mutations expected to result in loss or strong defect of the function of clathrin or of its endocytic adaptor AP-2 were shown not to abolish synaptic function (*Gu et al., 2008, 2013*; *Sato et al., 2009*). Additionally, studies of worm synapses expressing channelrhodopsin and subjected to a very brief photostimulus revealed an ultrafast endocytic reaction mediated by uncoated invaginations larger than SVs (*Watanabe et al., 2013a*). More recently, similar results – an ultrafast endocytic reaction mediated by large uncoated invaginations in response to a single optogenetic stimulus – were observed at synapses of mouse hippocampal neurons in primary culture (*Watanabe et al., 2013b*).

Both the molecular mechanisms underlying bulk endocytosis and those through which SVs are generated from bulk endosomes remain poorly understood. Like any other form of endocytosis, bulk endocytosis involves membrane remodeling and membrane fission. Thus, several studies of bulk endocytosis have addressed the potential involvement of dynamin, a GTPase known to mediate endocytic membrane fission in multiple contexts including, most prominently, endocytic fission at neuronal synapses (*Koenig and Ikeda, 1989*; *Ramaswami et al., 1994*; *Ferguson and De Camilli, 2012*). However, conflicting results have been reported. A role for dynamin, and more specifically for dynamin 1 has been supported by some studies (*Clayton et al., 2009, 2010*; *Xue et al., 2011*; *Nguyen*

et al., 2012). Interestingly, dynamin 1 is constitutively phosphorylated in resting nerve terminals and its $Ca^{2+}$ triggered dephosphorylation upon synaptic stimulation results in its binding to the F-BAR domain containing protein syndapin/pacsin (*Anggono et al., 2006*; *Clayton et al., 2009*; *Koch et al., 2011*), which has also been implicated in bulk endocytosis (*Andersson et al., 2008*). However, studies of dynamin 1 knock-out (KO) neurons have indicated the occurrence of robust bulk endocytosis even in the absence of dynamin 1 (*Hayashi et al., 2008*), in spite of a strong impairment of CME (*Ferguson et al., 2007*; *Lou et al., 2008*). It remains possible that other dynamins, dynamin 3 in particular, which is the other predominantly neuronal dynamin, may substitute for dynamin 1 in these neurons (*Ferguson et al., 2007*; *Raimondi et al., 2011*). Drugs (dyngo-4a and dynasore) that impair dynamin activity were reported to block bulk endocytosis (*Nguyen et al., 2012*). However, the interpretation of such results is questioned by the more recent demonstration that these drugs can robustly affect plasma membrane dynamics by dynamin-independent mechanisms (*Park et al., 2013*).

Concerning the conversion of 'bulk' endosomes into SVs, precise mechanisms have yet to emerge. Based on a study of 'broken' synaptosomes it was proposed that such a conversion occurs via the same clathrin-mediated budding reaction that drives clathrin-mediated budding from the plasma membrane (*Takei et al., 1996*). However, in that study bulk endosomes were exposed to conditions that can induce ectopic production of $PI(4,5)P_2$ (incubation with ATP and GTP or GTPγS) (*Seaman et al., 1993*; *Takei et al., 1996*; *Krauss et al., 2003*). Subsequent evidence that assembly of endocytic coats critically requires $PI(4,5)P_2$ in the membrane from which they originate (*Cremona et al., 1999*; *Höning et al., 2005*; *Zoncu et al., 2007*; *Idevall-Hagren et al., 2012*) challenged these results as $PI(4,5)P_2$, which is selectively concentrated at the plasma membrane, is generally rapidly depleted from endocytic membranes and is thus not expected to be concentrated on bulk endosomes (*Cremona et al., 1999*; *Chang-Ileto et al., 2011*; *Milosevic et al., 2011*).

The goal of this study was to gain new information into the recycling of SV membranes via bulk endocytosis. To this aim we examined the impact of manipulations that affect dynamin-dependent endocytosis on this process using electron microscopy (EM) in combination with endocytic tracers. We demonstrate that the rapid, bulk endocytic recapture of SV membranes induced by a strong stimulus, either KCl or high frequency stimulation, is not impaired at synapses lacking both of the two predominant neuronal dynamins, dynamin 1 and dynamin 3, that is synapses where CME is severely perturbed (*Ferguson et al., 2007*; *Raimondi et al., 2011*; *Ferguson and De Camilli, 2012*). Conversion of bulk endosomes into new SVs does occur in these synapses, although less efficiently than in wild-type (WT) synapses. The partial impairment of this conversion is not further aggravated by the additional knock-down (KD) of clathrin, indicating the clathrin-independence of this process. These observations provide evidence for a pathway of SV reformation that predominates after strong stimulation and does not require clathrin mediated budding or dynamin 1 and 3.

## Results

### Robust bulk endocytosis at dynamin 1 and 3 double KO synapses following high K+ stimulation

A previous study had suggested that stimulation of SV endocytosis by high extracellular $K^+$-induced depolarization results in a prominent burst of bulk endocytosis even in the absence of dynamin 1 (the major neuronal dynamin isoform, which is encoded by the *DNM1* gene) (*Hayashi et al., 2008*). To assess the contribution to bulk endocytosis of the other neuronal dynamin, dynamin 3, which is encoded by the *DNM3* gene, and to monitor reformation of SVs from bulk endosomes, we performed experiments with primary cultures of WT, dynamin 1 KO (Dyn1 KO), dynamin 3 KO (Dyn3 KO) and dynamin 1 and 3 double KO (Dyn1/3 DKO) mice. In order to reduce the contribution of genotypic variability, comparisons were made between cultures derived from littermates: (1) Dyn1 KO mice and WT mice within the litters obtained from the breeding of *DNM1+/−* mice and 2) Dyn1/3 DKO mice and Dyn3 KO mice within the litters obtained from the breeding of *DNM1+/−; DNM3−/−* mice.

Spontaneous network activity drives the accumulation of endocytic clathrin coated pits (CCPs) in cultures of Dyn1 KO or Dyn1/3 DKO neurons (*Ferguson et al., 2007*; *Hayashi et al., 2008*; *Raimondi et al., 2011*). However, silencing neuronal activity by an overnight incubation with the $Na^+$ channel blocker tetrodotoxin (TTX, 1 µM) nearly completely reverses this phenotype, as lack of dynamin 1 and 3 only severely delays, but does not completely block CME (*Raimondi et al., 2011*). Thus, in order to minimize functional and ultrastructural variability among synapses due to pre-existing activity, neuronal cultures

were pre-treated overnight with TTX, briefly rinsed and then washed for 5 min in buffer without TTX (and devoid of $Ca^{2+}$ to prevent SV exocytosis) prior to stimulation. Cultures were then stimulated for 90 s with 90 mM KCl to evoke maximal SV exocytosis and rapidly washed and further incubated in resting buffer (see below) for the analysis of SV recovery. To visualize endocytic organelles, cholera toxin-horseradish peroxidase (CTX-HRP) was added to the cultures both during stimulation and during the recovery. This probe binds the ganglioside GM1 on the cell surface and is an efficient marker of endocytic membranes (*Ferguson et al., 2007*; *Raimondi et al., 2011*).

Analysis of samples fixed at the end of the 90 s stimulation revealed that in WT synapses the number of SVs per synapse (number per $\mu m^2$ of synapse cross-section) was reduced to about 20% of the starting value (*Figures 1A and 2A*). Such reduction was accompanied by a massive appearance of endocytic vacuoles much larger than SVs, that is bulk endosomes (as defined by a diameter greater than 80 nm), nearly all of which were positive for CTX-HRP, as assessed by HRP cytochemistry (*Figures 1A and 2A,C*). Similar changes were observed in Dyn3 KO synapses (*Figure 1A*, *Figure 2B,D*). In contrast, in both Dyn1 KO synapses and Dyn1/3 DKO synapses, SVs were nearly completely depleted at the end of the stimulus (*Figures 1A, 2A,B*), indicating that SV reformation is blocked in these genotypes during stimulation. This difference was accompanied by a greater number of large endocytic interme-diates (diameter >80 nm) in these genotypes, relative to WT and to Dyn3 KO synapses (*Figures 1A and 2B,D*). Importantly, these findings demonstrate that bulk endocytosis is not impaired, and perhaps is even enhanced, in the absence of both Dyn1 and Dyn3.

## Reformation of SVs from bulk endosomes as revealed by CTX-HRP labeling

Reformation of SVs following interruption of the stimulus was monitored in either $Ca^{2+}$ free buffer or TTX in Tyrode buffer, two conditions that block further exocytosis and that pilot experiments revealed to result in a similar recovery of SVs. Following the stimulus, a very robust recovery of SV number was already observed after 10 min of recovery in WT and Dyn3 KO synapses (*Figures 1A and 2A,B*), with a corresponding decrease of HRP reaction product-positive large vacuoles (*Figure 2C,D*), and this recovery continued thereafter (*Figures 1A, 2A–D*). Recovery (both an increase in SV number and a decrease in larger endocytic intermediates) was also observed in Dyn1 KO and Dyn1/3 DKO synapses, although both SV reformation and endocytic vacuole clearance were delayed relative to littermate controls (WT and Dyn3 KO respectively; *Figure 2A–D*). Note that the overall similar results for WT and Dyn3 KO cultures and for Dyn1 KO and Dyn1/3 DKO cultures, confirm previously reported evidence for the predominant role of dynamin 1 in SV recycling.

In all samples exposed to this experimental protocol, the majority of newly reformed SVs were labeled (dark portions of each bar in *Figure 2*). Interestingly, even at the end of the 90 s stimulation, the majority of the remaining SVs in WT and Dyn3 KO synapses were labeled, indicating that they had formed by endocytic recycling during the stimulus (*Figure 1*, *Figure 2A,B*). As SVs had been nearly depleted in Dyn1 KO and Dyn1/3 DKO synapses at this time point (*Figure 2A,B*), one can conclude that in all four genotypes the high $K^+$ stimulus induces the exocytosis of all SVs, but that only at synapses containing dynamin 1 SV reformation occur during the stimulus, thus partially counteracting the loss of SVs (see also *Ferguson et al., 2007*; *Raimondi et al., 2011*). The occurrence of newly reformed CTX-HRP reaction product-negative SVs after 10 and 30 min of recovery in synapses that lack Dyn1 (*Figure 2A,B*) was surprising. In principle, nearly every new SV formed after the stimulus should be peroxidase reaction product positive as the contribution of new vesicle membrane delivered from the cell body is expected to be minimal over the time course of these experiments. Such occurrence may be explained by the stochastic nature of SV labeling and/or the all-or-none nature of the formation of peroxidase reaction product.

The conversion of bulk endosomes to SVs could in principle occur by two mechanisms. Bulk endosomes could be the direct source of SVs, or they may first fuse back to the plasma membrane from which new SVs could then bud by CME. To discriminate between these possibilities, cultures of all four genotypes were exposed to CTX-HRP only during the recovery phase, following a 90 s stimulus, and then analyzed at the 10 and 30 min time points (*Figure 1B*, *Figure 2E–H*). In these samples, only very few large vacuoles were HRP labeled (*Figure 2G,H*), indicating that bulk endocytosis occurs nearly exclusively during the high $K^+$ stimulation. If the conversion of bulk endosomes to SVs during stimulation requires their prior fusion with the plasma membrane, using this protocol the majority of newly reformed SVs should be labeled. This was not the case, as the fraction of CTX-HRP labeled

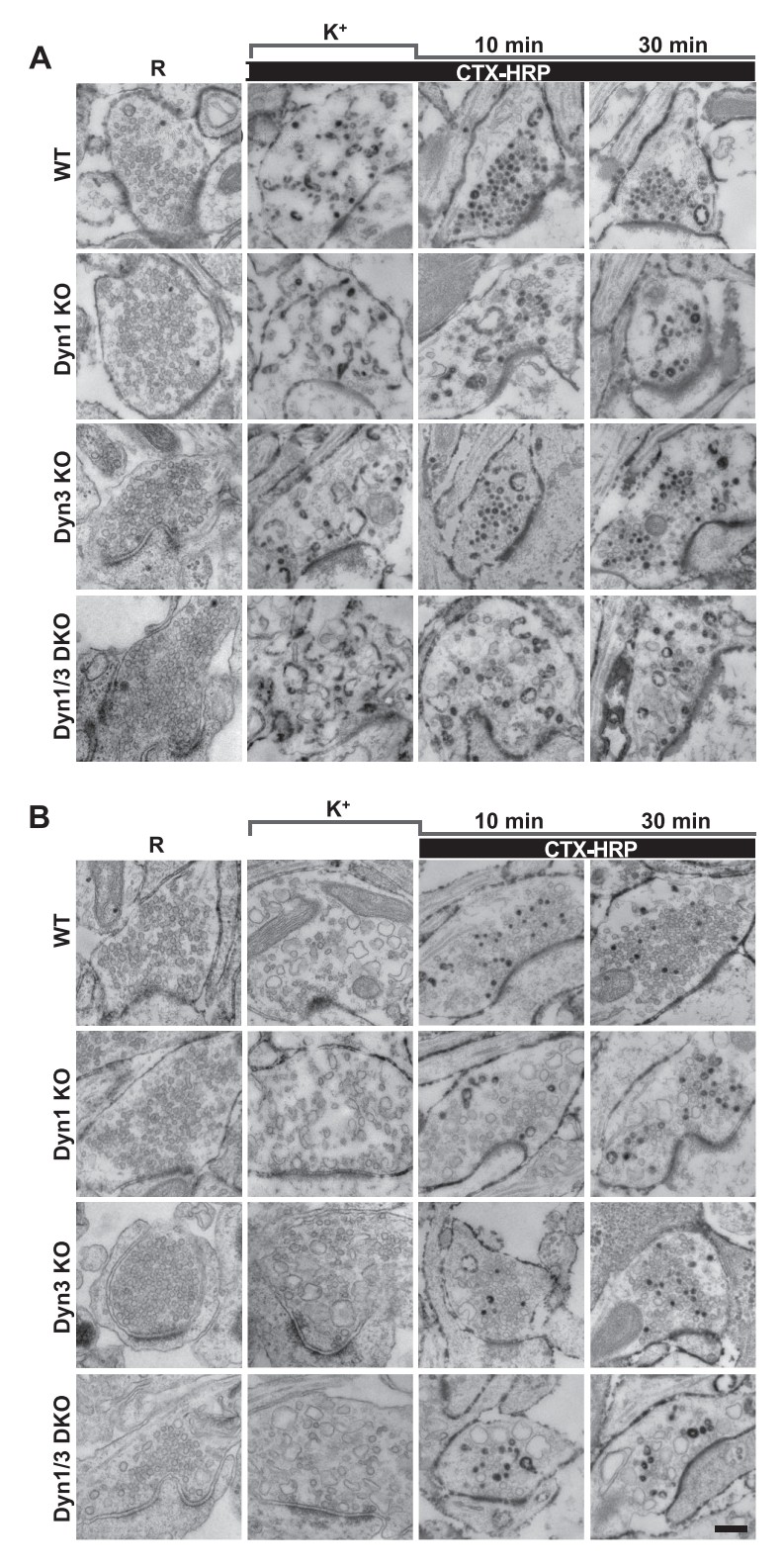

**Figure 1**. High K+ stimulation induces bulk endocytosis, as assessed by CTX-HRP, at both WT and dynamin mutant synapses. Following incubation in resting buffer (R), neurons were stimulated with 90 mM K+ for 90 s, then incubated in resting buffer for 10 and 30 min. CTX-HRP, added either during or after the stimulus (**A**), or selectively after the stimulus (**B**), was used as an endocytic tracer. In all genotypes, loss of SVs during the stimulus is compensated

*Figure 1. Continued on next page*

*Figure 1. Continued*

by the appearance of numerous bulk endosomes. Most these organelles are HRP-reaction product-positive only in samples expose to CTX-HRP also during the high K$^+$ incubation, indicating that bulk endocytosis occurs primarily during the stimulus. During the recovery period bulk endosomes are converted to SVs. Scale bar = 100 nm.

newly reformed SVs, relative to all SVs, was much lower than in samples incubated with CTX-HRP both during and after the stimulus (compare the solid to empty portions of the bars in *Figure 2E,F* and in *Figure 2A,B*, respectively, see also *Figure 2I*). These findings imply that during recovery a large pool of SVs reforms directly from the bulk endosomes that were generated during the stimulus. Note that the number of CTX-HRP positive SVs, that is SVs generated during post-stimulus endocytosis, was only modestly decreased in Dyn1 KO and Dyn1/3 DKO synapses relative to their controls (*Figure 2E,F*), suggesting that either clathrin/dynamin-dependent SV endocytosis is not involved in their formation or that dynamin/clathrin-mediated endocytosis occurring during the recovery can be sustained by dynamin 2 even though the levels of this dynamin isoform in neurons are very low compared to either dynamin 1 or 3 (*Ferguson et al., 2007*; *Raimondi et al., 2011*).

From the analysis of the data presented so far, one can draw the conclusions summarized in the pie charts of *Figure 2I* concerning the reformation of SVs at 10 and 30 min following the high K$^+$ stimulus. For each condition, the entire pie (calculated by the data of fields A and B) represents the global SV pool that is present in nerve terminals before the stimulus, and which undergoes massive exocytosis with the stimulation. The sector delineated by a dotted line reflects the pool of SVs that was still missing at the time point of recovery examined, revealing a delay in SV reformation in the absence of the neuronal dynamins. The white sector of the pie represents SVs that are unlabeled by CTX-HRP. In spite of the absence of HRP reaction product, even these SVs are likely to be of endocytic origin given the complete depletion of SVs produced by the stimulus. As stated above, these vesicles are most likely unlabeled due to the less than 100% efficiency of labeling, as supported by the observation that the size of this pool roughly correlates with the number of recovered SVs. The black sector shows the CTX-HRP-labeled SVs reformed during the 90 s stimulus, while the striped sectors represent the labeled SVs generated during recovery (solid portions of the bars during recovery in fields A and B, subtracted by the solid portion of the K$^+$ bars). Within the striped sectors, the portion with red stripes represents the fractions of SVs that become labeled when CTX-HRP is added only after the stimulus (calculated from the solid portions of the bars of fields E and F), thus implying that the portion with black stripes represents SVs that form from bulk endosomes generated during the stimulus. Our conclusions from these comparisons are as follows: (1) SV reformation occurs even in the absence of Dyn1 and Dyn3, but this process is substantially delayed as shown by the missing portion of the pie; (2) No SV reformation occurs during the stimulus in the absence of Dyn1 or both Dyn1 and 3 (solid black sections), leading to complete SV depletion; (3) Reformation of SVs by endocytosis that occurs after the interruption of the stimulus is surprisingly similar in all genotypes (red stripes sector); (4) Most of the delay in recovery in dynamin KO neurons is due to a delay in the conversion from bulk endosomes to SVs (black stripes).

## Fluid phase HRP labeling confirms direct budding of SVs from bulk endosomes

To further assess the delay in the generation of new SVs from bulk endosomes and the consequences of the absence of dynamin 1 and 3 on this process, further experiments were performed that employed fluid phase HRP (rather than CTX-HRP) to follow SV recycling. While this probe labels endocytic vesicles less efficiently than CTX-HRP, as it is not bound to, and therefore concentrated at, membranes, it can be rapidly washed away from the cell surface, and thus can be used to better analyze endocytosis in a specific time window. Neuronal cultures were prepared and stimulated as described above, but with the presence of free HRP during the stimulation and with the immediate wash of HRP at the end of the stimulation.

As observed in experiments with CTX-HRP, SV number was strongly decreased, but not abolished in WT and Dyn3 KO nerve terminals at the end of the stimulus (*Figures 3 and 4A,B*), when numerous HRP reaction product-positive vacuoles had accumulated (*Figure 4C,D*). Additionally, a fraction of such vesicles was HRP reaction product-positive, consistent with the occurrence of SV reformation even during stimulation (*Figures 3 and 4A,B*). Importantly, there was a further increase of HRP-labeled

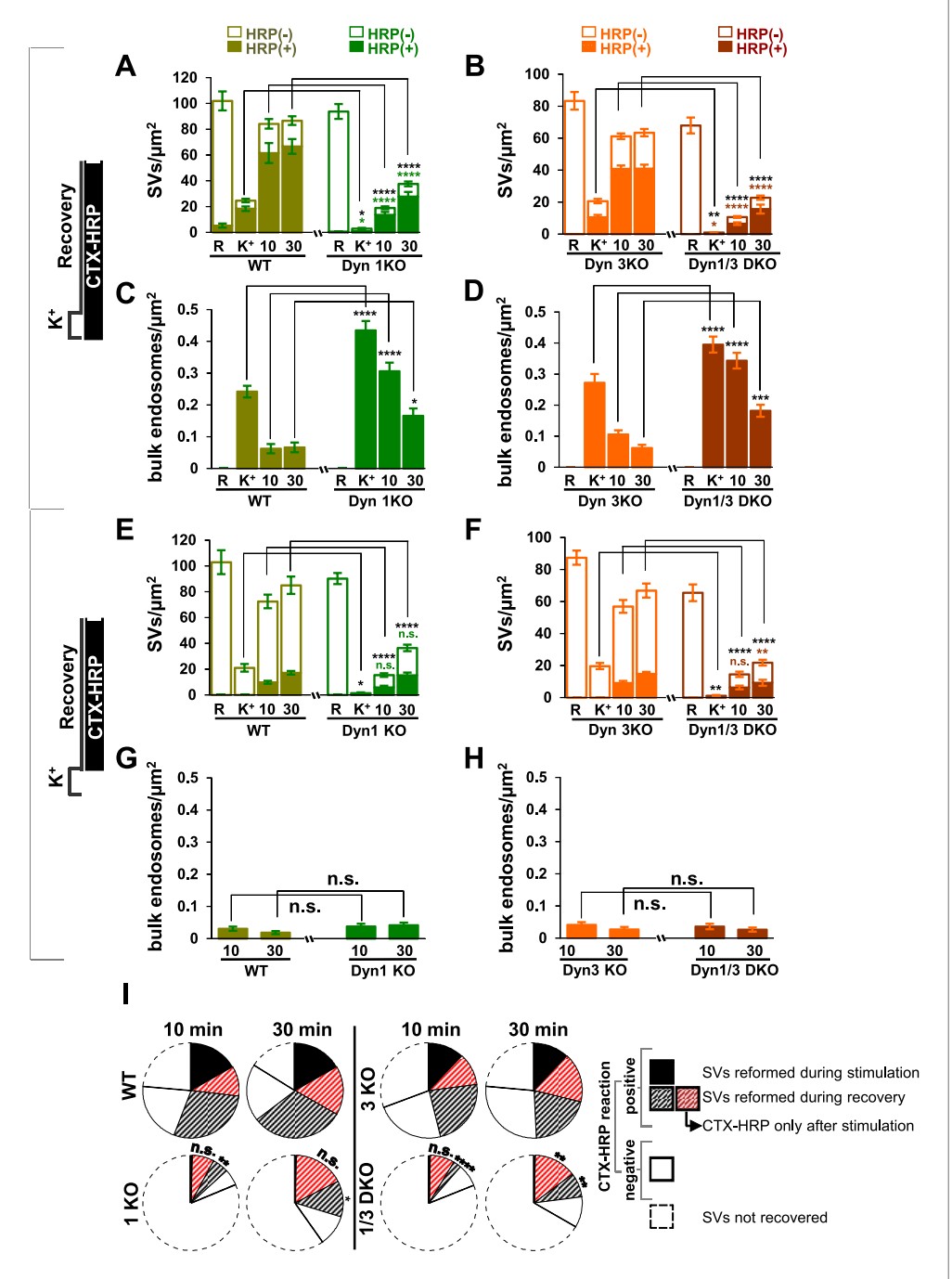

**Figure 2**. Quantification of SV number and bulk endosome dynamics, as assessed by CTX-HRP labeling, in WT and dynamin mutant neurons in response to high K⁺ stimulation. These data represent a quantification of the experiments used for *Figure 1*. CTX-HRP was present both during and after stimulation in (**A**–**D**), in order to maximize labeling of all endocytic structures, and only after stimulation in (**E**–**H**) to monitor selectively post-stimulation endocytosis. (**A**, **B**, **E** and **F**) Analysis of SV number per cross-sectional area of nerve terminal under resting conditions (R), at the end of the 90 s high K⁺ stimulus (K⁺), and after 10 and 30 min recovery. SVs positive for HRP reaction product are indicated by the dark portion of each bar. (**C**, **D**, **G** and **H**) Analysis of the membrane area of CTX-HRP-positive bulk endosomes per cross-sectional area of nerve terminal in each of the conditions investigated. In fields (**A**–**H**) as well as in the following figures, error bars represent mean ± S.E. of values obtained at individual synapses. (**I**) Pie charts integrating results of different experiments and summarizing putative contributions of (i) direct endocytosis from the plasma membrane and (ii) conversion of bulk endosomes to the recovery of

*Figure 2. Continued on next page*

*Figure 2. Continued*

SVs at 10 and 30 min as extrapolated from the data of fields (**A** and **B**) and (**E** and **F**). The entire pie represents the SV pool before stimulation. The black section of each pie indicates HRP-reaction positive SVs at the end of the stimulus. The striped sections indicate HRP-reaction positive SVs generated during recovery (dark portion of the 'recovery' bars in field **A** and **B**, subtracted by the dark portion of the 'K+' bars in the same fields), with red stripes showing SVs labeled when CTX-HRP had been added selectively after the stimulus (dark portions of the bars in fields **E** and **F**). As very few bulk endosomes are generated when CTX-HRP is added selectively after the stimulus (fields **E** and **F**), most likely these vesicles (red stripes) were derived directly from the plasma membrane. Thus, black striped sections represent SVs putatively derived from bulk endosomes. The white sections represent SVs unlabeled by HRP and the missing section represents the vesicle pool that had not recovered at the time points indicated. The pie clearly demonstrates a delay in SV recovery in the Dyn1 KO and Dyn1/3 DKO genotypes. It further shows that in these genotypes the predominant defect is in the reformation of SVs during the stimulus and from bulk endosomes, with no obvious impact on their reformation from the plasma membrane after stimulation. ****, ***, **, * indicate p-values of <0.0001, <0.001, <0.01 and <0.05, respectively. n.s., 'not significant'. Black asterisks refer to comparisons between total vesicles, colored asterisks to comparisons between HRP-labeled vesicles. Bars: standard error of the mean.

SVs during the recovery in the HRP-free medium, and a corresponding decrease of larger vacuoles, indicating a contribution of HRP reaction product-positive vacuoles formed during the stimulus to SV reformation (*Figures 3 and 4A,C*). In Dyn1 KO and Dyn1/3 DKO synapses, SVs were depleted at the end of the stimulus (*Figures 3 and 4A,B*) and the accumulation of vacuolar intermediates (HRP labeled) was larger than in controls (*Figures 3 and 4C,D*). In the post-stimulation period, SVs reformed and vacuoles were consumed, but with some delay relative to controls (*Figure 4A*) consistent with results of experiments with CTX-HRP. Collectively, these experiments demonstrate that neither Dyn1 nor Dyn3 is required for bulk endocytosis and that their absence delays, but does not abolish, SV reformation from internal membranes.

Results with fluid phase HRP are summarized in the pie charts of *Figure 4E,F*, which shows SV number at various time points. Sectors delimited by a dotted line reflect the SV pool that did not reform, while white sectors delineated by a continuous black line indicate unlabeled SVs (the fraction of unlabeled vesicles is higher than with CTX-HRP as the endocytic probe is not pre-enriched at the membrane). The black sectors represent the HRP-labeled SVs that reformed during the stimulus, when extracellular HRP was present (solid portions of the high K+ bars in *Figure 4A,B*), while the striped sectors indicate the additional HRP-labeled SVs formed during recovery when extracellular HRP was absent (solid portions of the bars during recovery subtracted by solid portion of the high K+ bars), and thus derived from bulk endosomes. As in the case of experiments with CTX-HRP, one can conclude that SV reformation is blocked during stimulation and delayed during recovery, with at least a portion of this delay being explained by a delay in the conversion of bulk endosomes to SVs.

## HRP-labeled recycled vesicles can undergo a new round of secretion

It was important to show that vesicles labeled by the endocytic tracer are indeed *bona fide* SVs, that is that they are competent for a new cycle of stimulated secretion. To address this issue, Dyn1/3 DKO cultures, and Dyn3 KO cultures as controls, were exposed to high K+ stimulation in the presence of soluble HRP, then washed and further incubated for 30 min in resting medium in the absence of HRP, exposed to a second high K+ stimulus, and finally rested again for 30 min, always in the absence of HRP. As shown by EM analysis at the end of each period (*Figure 5*), the second stimulus was as effective as the first in inducing exocytosis of both labeled and unlabeled SVs.

## Results obtained with high K+ depolarization can be replicated by high frequency electrical stimulation

Synapse stimulation with high K+-induced depolarization for 90 s offered the opportunity to monitor reformation of SVs after their massive depletion. However, depolarization by high K+ represents a non-physiological stimulus. Thus, to assess the physiological relevance of our results, we performed a morphological analysis on neuronal cultures from Dyn3 KO and Dyn1/3 DKO synapses exposed to a 10 s electrical stimulation at 80 Hz (*Wenzel et al., 2012*) and then allowed to recover for 30 min. For this analysis, the CTX-HRP labeling protocol was used: cultures were incubated with CTX-HRP both during stimulation and during recovery, or during the recovery only (*Figure 6A,B*).

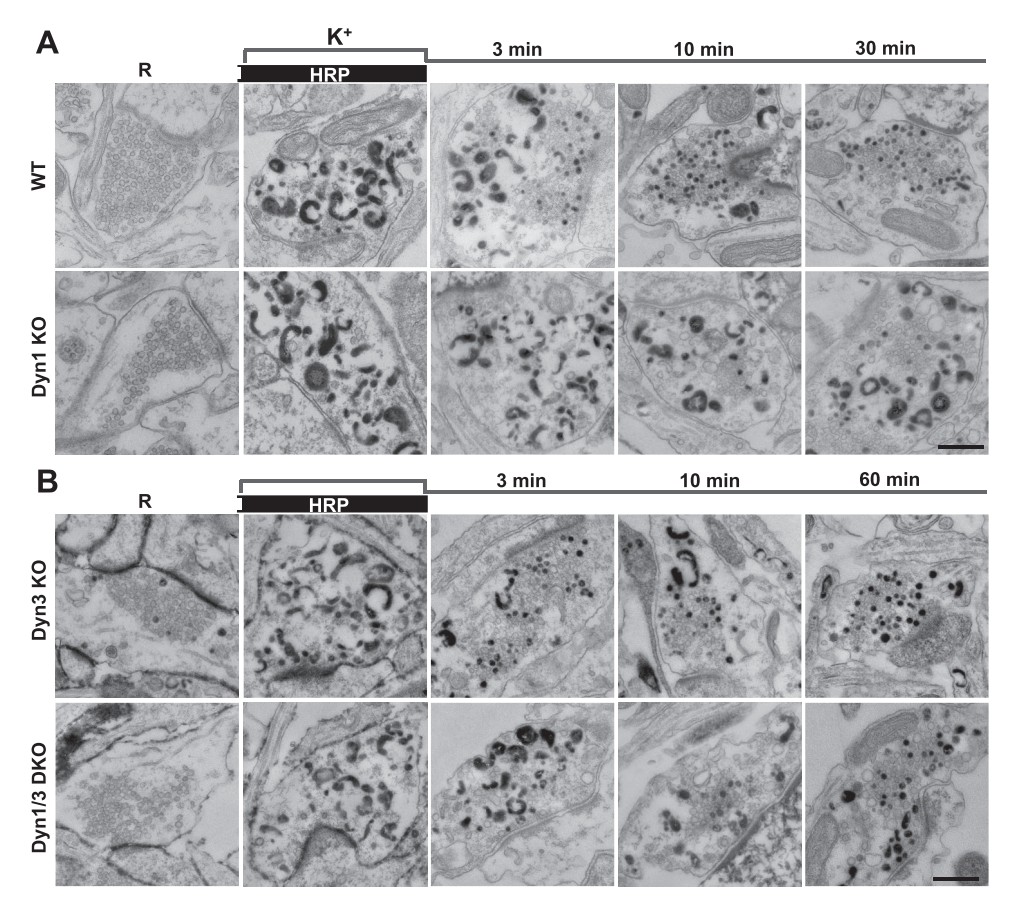

**Figure 3**. Bulk endocytosis and conversion of bulk endosomes into SVs analyzed with soluble HRP labeling after high $K^+$ stimulation in WT and dynamin mutant synapses. Neurons of the indicated genotypes (**A**, WT and Dyn1 KO; **B**, Dyn3 KO and Dyn1/3 DKO respectively) were stimulated with 90 mM $K^+$ for 90 s in the presence of HRP, followed by wash-out of HRP and recovery in resting buffer. Massive bulk endocytosis is observed during stimulation. The large fraction of HRP reaction product-positive SVs observed in all genotypes during the recovery period documents formation of these vesicles from HRP reaction product-positive bulk endosomes formed during stimulation. Scale bar = 500 nm.

80 Hz electrical stimulation yielded results very similar to those produced by high $K^+$ stimulation. A robust reduction in SV number was observed immediately after stimulation, and this effect was stronger at Dyn1/3 DKO synapses. Even in this case, when CTX-HRP labeling was present during stimulation, some labeled SVs were observed at the end of the stimulus, but only in the control (Dyn3 KO) cultures (*Figure 6A,B*, and black sector in the pie chart of *Figure 6C*). Furthermore, the percentage of recovered vesicles and the fraction of CTX-HRP-labeled vesicles at the end of recovery in both labeling modes and for both genotypes, was similar to those observed for high $K^+$ stimulation (*Figure 6A,B*). As the pie chart of *Figure 6C* shows (labeling of the sectors is the same as in *Figure 2I*), the strongest defects observed at Dyn1/3 DKO synapses relative to controls concern the reformation of SVs during the stimulus (black sector) and the recovery of SVs from CTX-HRP labeled bulk endosomes generated during the stimulus (sector with black stripes).

## Clathrin-independent budding from bulk endosomes

Given evidence for a robust contribution of bulk endosomes to the reformation of SVs, the potential role of clathrin to the budding of new SVs from such organelles during recovery from an acute stimulation was explored. CCPs were nearly absent before stimulation in synapses of all genotypes, including Dyn1 KO and Dyn1/3 DKO synapses (*Figure 7A–D*), as spontaneous network activity prior to the stimulation had been silenced by an overnight treatment with TTX.

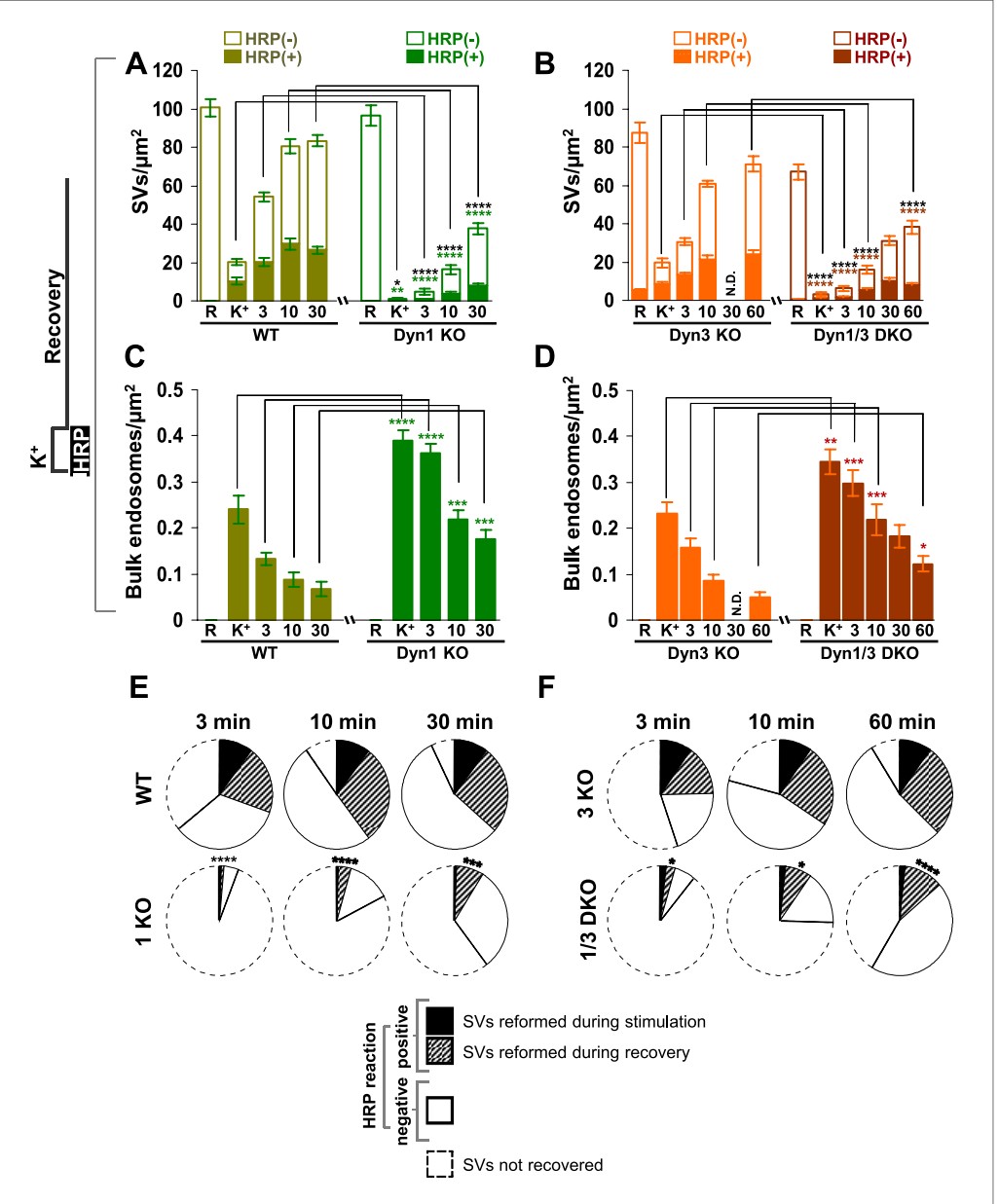

**Figure 4**. Quantification of SV number and bulk endosome dynamics, as assessed by soluble HRP labeling, in WT and dynamin KO neurons in response to high K+ stimulation. These data represent a quantification of the experiments used for *Figure 3*. (**A** and **B**) Analysis of SV number per cross-sectional area of nerve terminal under resting conditions (R), at the end of the 90 s high K+ stimulus (K+), and after recovery for the times indicated. SVs positive for HRP reaction product are indicated by the dark portion of each bar. (**C** and **D**) Analysis of the membrane area of HRP reaction product-positive bulk endosomes per cross-sectional area of nerve terminal in each of the conditions investigated. (**E** and **F**) Pie charts summarizing results of fields **A** and **B** concerning the reformation of SVs at different times after recovery. Entire pie: total SVs before stimulation; white sector: unlabeled SVs; black sector: labeled SVs after stimulation; striped section: total labeled SVs subtracted by the SVs formed during stimulation. ****, ***, **, * indicate p-values of <0.0001, <0.001, <0.01, and <0.05, respectively. N.D., 'not determinated'. Black asterisks refer to comparisons between total vesicles, colored asterisks to comparisons between HRP-labeled vesicles. Bars: standard error of the mean.

Upon stimulation, CCP number increased only slightly in WT and Dyn3 KO synapses, but more strongly in Dyn1 KO and Dyn1/3 DKO synapses (*Figure 7A–D*), where numerous CCPs located deep inside nerve terminals were observed. Such CCPs were accessible to CTX-HRP added during recovery,

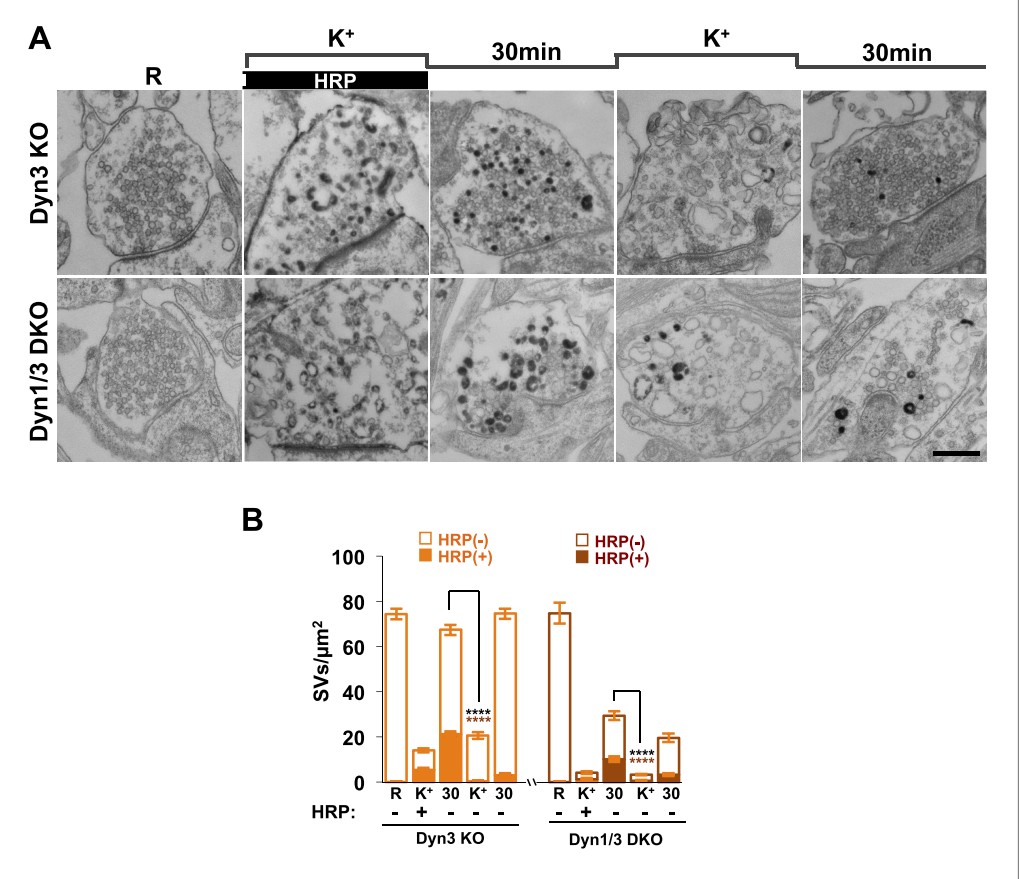

**Figure 5**. SVs reformed from bulk endosomes can undergo exocytosis upon stimulation. (**A**) Control (Dyn3 KO) and Dyn1/3 DKO neurons were stimulated by a 90 s high K+ stimulation in the presence of HRP, rested for 30 min in the absence of HRP and then subjected to a second round of high K+ stimulation and recovery, always in the absence of HRP. Loss of HRP-labeled SVs during the second stimulus, and the subsequent recovery of unlabeled SVs, demonstrates that vesicles generated from bulk endosomes are functional SVs. (**B**) Quantification of the results shown in **A**. Scale bar = 500 nm. **** indicates p-values of <0.0001. Black asterisks refer to comparisons between total vesicles, colored asterisks to comparisons between HRP-labeled vesicles. Bars: standard error of the mean.

as reflected by the presence of HRP reaction product in their core (*Figure 7A,B,E*). If these CCPs represent buds on bulk endosomes, at least some of them should also become HRP reaction product-labeled in samples incubated with fluid-phase HRP during the 90 s stimulation and then immediately washed. Under these conditions, only CCPs on endosomal membranes, but not plasma membrane-connected CCPs, are expected to exhibit HRP signal. However, in these samples, CCPs were observed only on HRP reaction product-negative membranes (*Figure 7B,D,F*). These results indicate that CCP formation in response to high K+ stimulation predominantly occurs on the plasma membrane and not on endocytic intermediates formed by bulk endocytosis.

EM tomography further confirmed that CCPs present deep inside terminals were connected to the plasma membrane. 3D models from the tomographic reconstruction of WT and Dyn1 KO synapses incubated with fluid-phase HRP during the stimulus and then allowed to recover for 1 min are shown in *Figure 8A–D*. SVs (blue) were strongly depleted at the Dyn1 KO synapse. No CCPs (green) were observed on bulk endosomes (yellow, identified by an HRP reaction product-positive core and anatomical separation from the plasma membrane) in either WT or Dyn1 KO. Furthermore, the overall appearance and distribution of HRP reaction product-positive vacuoles (yellow) was similar in the two genotypes: irregular cisternae, which in some cases had tubulated portions. (*Figure 8A,C* and selected examples in *Figure 8E,F*). CCPs were frequently found in Dyn1 KO synapses and massively abundant at a subset of Dyn1 KO synapses, but in these synapses they were selectively localized on deep,

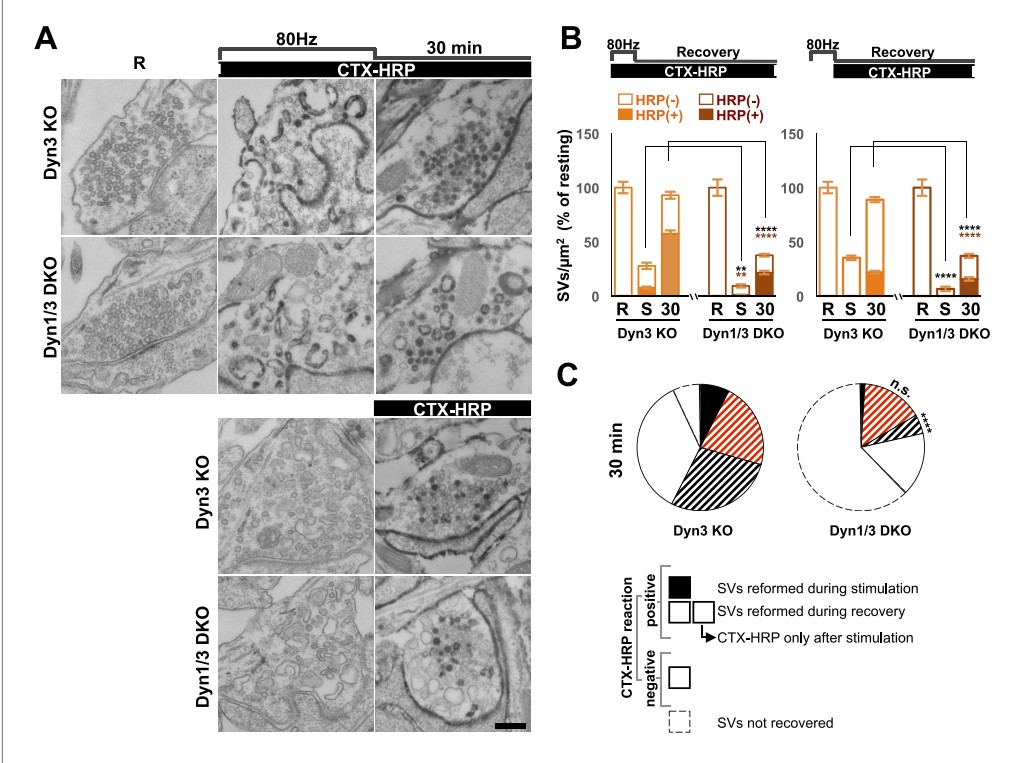

**Figure 6**. Bulk endocytosis and SV recovery after high frequency electrical stimulation. (**A**) Neuronal cultures were stimulated at 80 Hz for 10 s and then allowed to recovery in resting buffer for 30 min. CTX-HRP was added either during both stimulation and recovery, or during recovery only, as indicated. The endocytic intermediates observed under these conditions, as well as their labeling pattern, were similar to those observed upon high K$^+$ stimulation. (**B**) quantification of the results shown in **A**. (**C**) Pie charts illustrating SVs observed under the various conditions, based on the results shown in panel **B**, with sectors coded as indicated in the legend of *Figure 2I*. As in the case of high K$^+$ stimulation (*Figure 2*), SV reformation during the stimulus was virtually abolished in Dyn1/3 DKO neurons (black sectors). Concerning the recovery of SVs after the stimulus (striped sectors), the major defect was observed in the reformation of labeled SVs from bulk endosomes (black stripes), this value was extrapolated (as for *Figure 2I*) by subtracting from the total striped area the fraction of SVs that become labeled when CTX-HRP was added only after the stimulus. Scale bar = 100 nm. ****, ** indicate p-values of <0.0001 and <0.01, respectively. n.s., 'not significant'. Black asterisks refer to comparisons between total vesicles, colored asterisks to comparisons between HRP-labeled vesicles. Bars: standard error of the mean.

narrow invaginations of the plasma membrane (*Figure 8D*) (a connection between one such invagination and the plasma membrane is shown in the inset of *Figure 8D*).

To assess in more detail the potential occurrence of budding intermediates on bulk endosomes under conditions that optimally preserve their native ultrastructure, we performed high pressure freezing/freeze-substitution on neuronal cultures from Dyn1/3 DKO mice and from their control (cultures from Dyn3 KO siblings) cultured on sapphire disks and exposed to resting, stimulatory (90 s high K$^+$) and recovery (3 and 10 min) conditions.

EM analysis of these preparations revealed a more compact texture of the neuronal cytoplasm and a different view of endocytic structures induced by stimulation relative to conventionally fixed material. The great majority of bulk endosomes, which had an electron-lucent lumen, had smoother rounded or oval-shaped profiles (*Figure 9A*, see also tomographic reconstructions at 3 min recovery in *Figure 9—figure supplement 1*). This was in contrast to the highly irregular shapes of bulk endosomes observed in conventionally fixed specimens, even more so when they were filled by HRP reaction product (*Figures 1 and 3*). Only occasionally (WT: 8 of 275; Dyn1 KO: 15 of 249; Dyn3 KO: 8 of 235; Dyn1/3 DKO: 32 of 258 synapses) were coated buds observed on vacuolar structures (*Figure 9B*) and, with only one exception, when the entire profile of these structures was re-constructed within

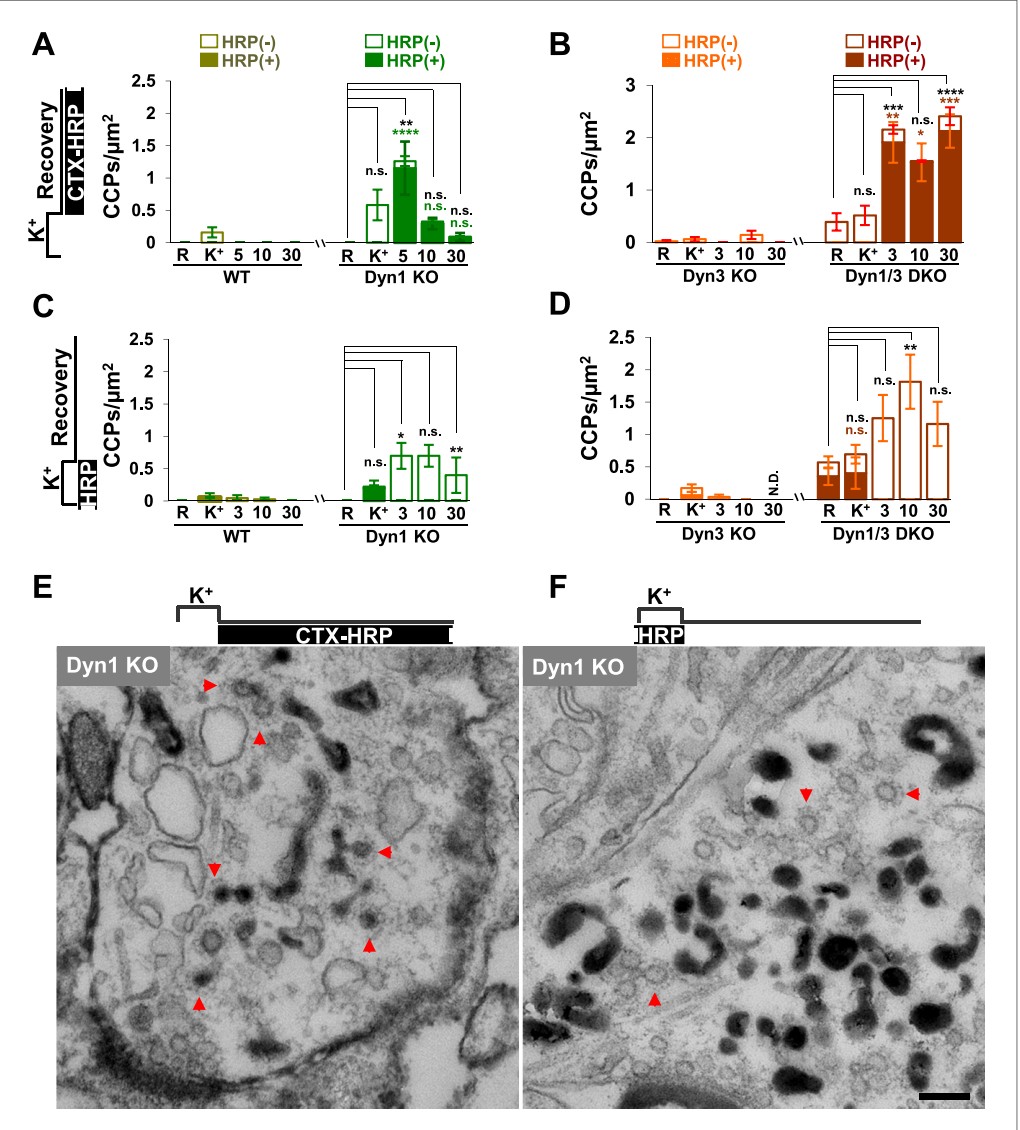

**Figure 7**. CCP number following stimulation is drastically elevated at dynamin mutant synapses, but in both WT and KO synapses they are selectively localized in the plasma membrane. (**A** and **B**) CCP number in cultures incubated with CTX-HRP selectively after the high K⁺ stimulation. CCPs positive for HRP reaction product are indicated by the dark portion of each bar. Note that in these samples, where only very few vacuoles are labeled (see *Figure 1G,H*), nearly all CCPs are labeled, indicating their localization on the plasma membrane. (**C** and **D**) CCP number in cultures stimulated in the presence of soluble HRP and washed at the end of the stimulus. Note that CCPs observed after stimulation are negative for HRP reaction product (dark portion of each bar), once again indicating their selective localization on the plasma membrane. The HRP-positive CCPs in the resting (R) and stimulated (K⁺) samples reflect accessibility to the extracellular medium of arrested plasma membrane CCPs in dynamin KO synapses. (**E**) Electron micrograph of a Dyn1 KO synapse incubated for 5 min with CTX-HRP selectively during recovery (as for field **A**). Most CCPs are HRP reaction product positive (red arrows). (**F**) Dyn1 KO synapse incubated with soluble HRP during stimulation and then washed and examined at 3 min (as for field **C**). All CCPs are HRP reaction product negative. Scale bar = 100 nm. ****, ***, **, * indicate p-values of <0.0001, <0.001, <0.01, and <0.05, respectively. n.s., 'not significant', N.D., 'not determined'. The standard error of the mean is shown in each graph. Black asterisks refer to comparisons between total vesicles, colored asterisks to comparisons between HRP-labeled vesicles. Bars: standard error of the mean.

tomograms, they were found to be continuous with the plasma membrane (*Figure 9B*, field B4 and *Figure 9C*, field C4). As expected, CCP 'trees' were only observed in recovered Dyn 1/3 DKO synapses (*Figure 9C*).

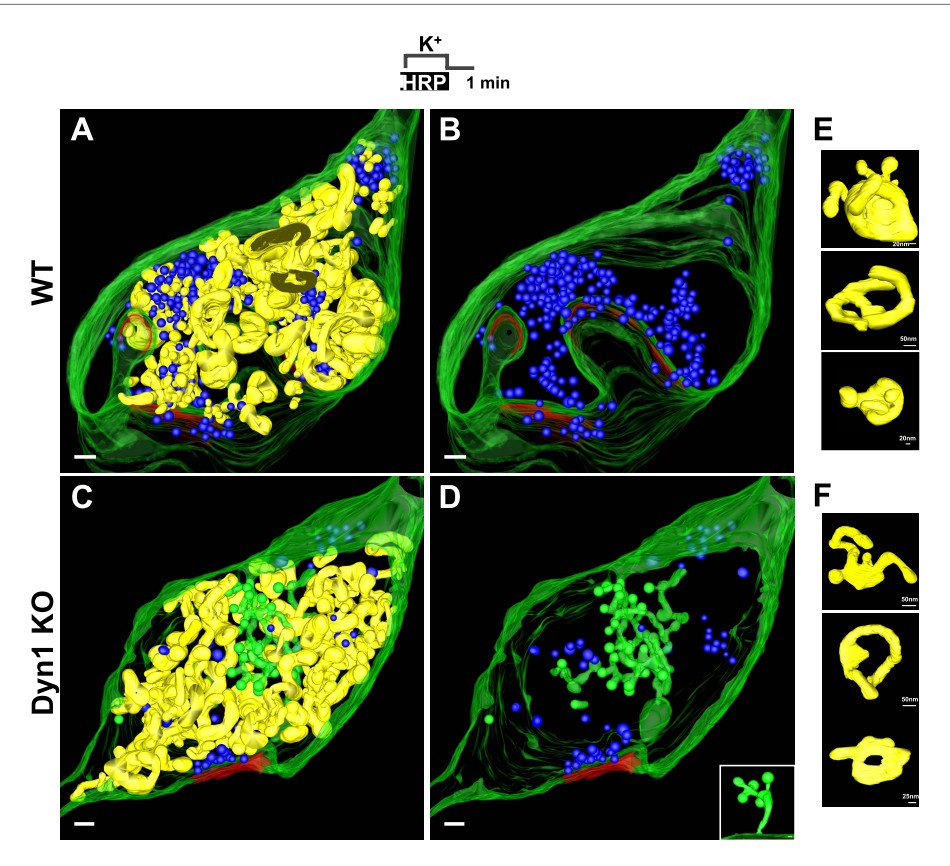

**Figure 8**. 3D models based on electron microscopy tomograms of WT and Dyn1 KO synapses following high K+ stimulation. The models illustrate the abundance and morphology of HRP-labeled bulk endosomes (yellow) in nerve terminals from WT (**A** and **B**, a 600 nm thick volume) and Dyn1 KO (**C** and **D**, an 800 nm volume) neuronal cultures stimulated for 90 s with high K+ in the presence of soluble HRP and recovered for 1 min in HRP-free buffer. PM is shown in green. **B** and **D** show the same nerve terminals of **A** and **B** after subtraction of bulk endosomes to reveal SVs (blue) and CCPs (green). CCPs are shown in the same green as the plasma membrane as they are all connected to this membrane by long narrow tubules (see a direct connection to the plasma membrane in the inset of **D**). No CCPs are present on bulk endosomes. The postsynaptic density is shown in red. Scale bars: **A**–**D**, 100 nm. (**E** and **F**) Gallery of individual bulk endosomes in WT and Dyn1 KO nerve terminals, respectively.

Tubulated, beaded extensions of some bulk endosomes were also visible in these preparations (*Figure 9—figure supplement 1*, *Figure 9D*), suggesting the potential occurrence of new SVs by progressive fragmentation. If this was the case, one would expect vesicles generated from bulk endosomes to be more irregular in size than vesicles derived from the classical CME at synapses, as CCPs and CCVs of nerve terminals typically have very homogeneous size. In fact, a morphometric analysis of thin sections from high-pressure frozen and freeze-substituted specimens revealed a narrow range of SV diameters that peaked at 30 nm at resting synapses (*Figure 10A,B*). After stimulation and 3 or 10 min recovery (*Figure 10A,B*) not only was there a very striking increase in the number of vesicles above 80 nm (our set point to define vesicles as bulk endosome), but a broader distribution of smaller vesicles with a general shift towards larger diameters was observed.

## Clathrin knock-down does not further impair SV reformation at Dyn1/3 DKO synapses

To directly address a potential clathrin independence of SV reformation form bulk endosomes, we performed clathrin heavy chain (CHC) KD experiments using a lentivirus based strategy (*Thomas et al., 2009*; *Ritter et al., 2013*). CHC was knocked down in Dyn1/3 DKO neurons by transducing them with lentivirus encoding CHC shRNAmiR or control shRNAmiR. Based on the analysis of GFP

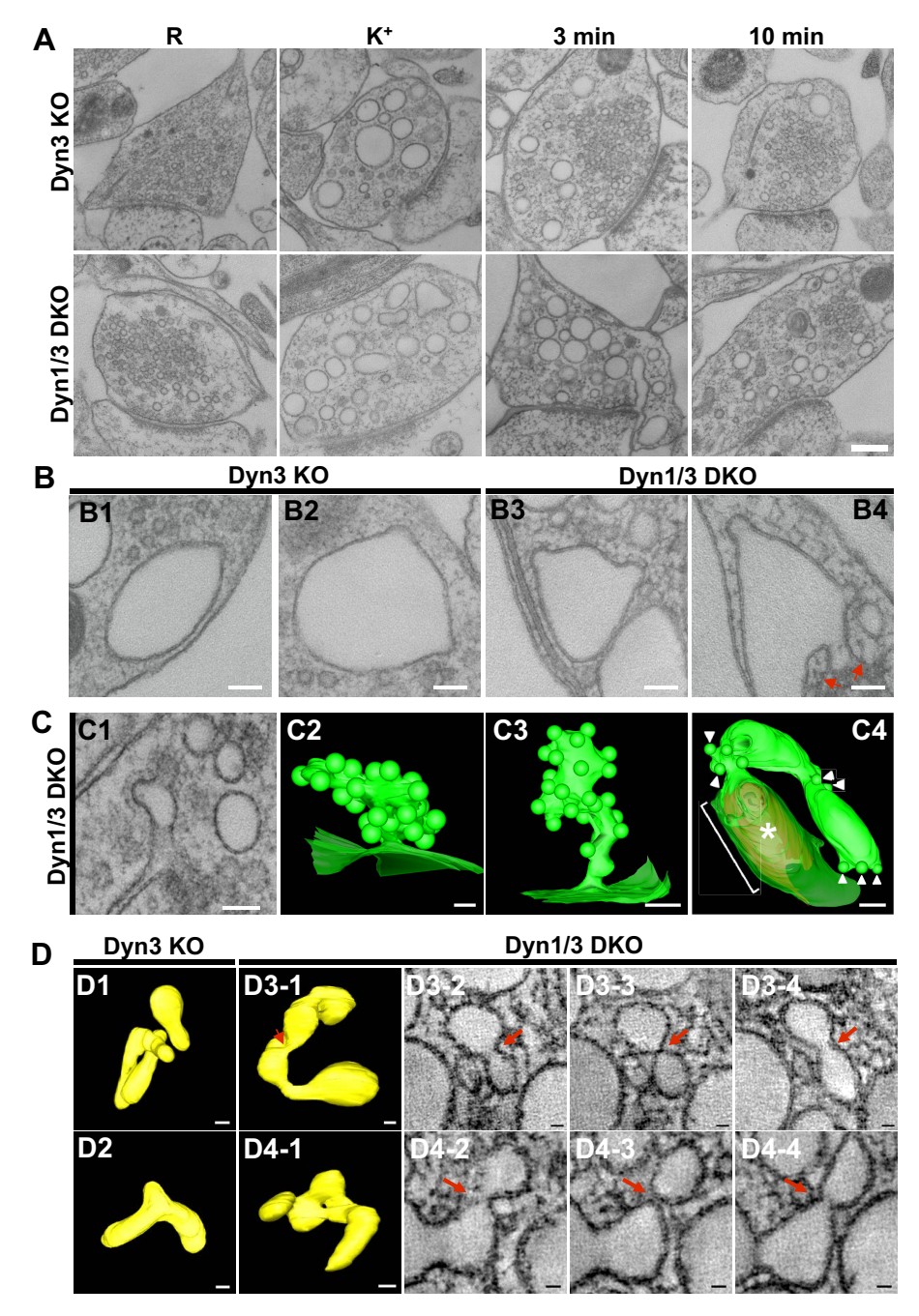

**Figure 9**. EM analysis of bulk endosomes from high pressure-frozen/freeze-substituted specimens. Neuronal cultures were incubated overnight in TTX and then exposed to 90 s stimulation in high K+ and recovered for 3 and 10 min. (**A**) Representative examples of Dyn3 KO (control) and Dyn1/3 DKO synapses. (**B**) Gallery of coated buds on vacuole-like structures: note the connection to the plasma membrane in B4 (red arrows). (**C**) CCPs on PM invaginations at Dyn1/3 DKO synapses. C1, EM micrograph; C2-C3, 3-D models from tomograms illustrating CCP trees; C4, CCPs (white arrowheads) on a dilated PM invagination. A white bracket and an asterisk indicates the portion of the PM directly apposed to the postsynaptic membrane (red) which is visible through the transparent green presynaptic plasma membrane. (**D**) Constricted bulk endosomes. Serial images D3-2 to D3-4 show constrictions and serial images D4-2 to D4-4 show constrictions and a fenestration. Red arrows point to membrane discontinuities. Scale bars: **A**, 250 nm; **B** and C1, 3 and 4, 100 nm; C2, 50 nm; **D**, 50 nm (3D), 20 nm (single images).

*Figure 9. Continued on next page*

*Figure 9. Continued*

The following figure supplements are available for figure 9:

**Figure supplement 1**. 3D models of high pressure-frozen/freeze substituted synapses from Dyn3 KO and Dyn1/3 DKO synapses at 3 min recovery after stimulation with high K⁺ for 90 s.

co-expressed by the shRNAmiR encoding virus, approximately 90% of the neurons were transduced and massive KD of CHC was confirmed by western blotting (*Figure 11A*). As expected, a dramatic reduction in CCP abundance in cultures of Dyn1/3 DKO neurons maintained under spontaneous levels of network activity was observed following CHC depletion (*Figure 11B*).

To test the contribution of clathrin to the recovery of SVs from bulk endosomes, CHC depleted Dyn1/3 DKO neurons (and Dyn1/3 DKO neurons as controls) were exposed to the same protocol used for *Figures 3 and 4* (overnight silencing with TTX, 90 s stimulation in HRP containing high K⁺ buffer and finally recovery in HRP-free resting medium) with analysis by EM. Prior to stimulation, only occasional CCPs were observed at synapses of both genotypes [overnight silencing of electrical activity nearly abolished CCPs, even in neurons not exposed to CHC KD (*Raimondi et al., 2011*)]. As expected, the increase in CCP number produced by the 90 s high K⁺ stimulation in Dyn1/3 DKO neurons was not observed in cells that had also been depleted of CHC (*Figure 11D*). However, the total recovery of SVs following 30 min of high K⁺ stimulation was similar at synapses of Dyn1/3 DKO neurons transduced with either the control or CHC KD conditions (*Figure 11C,E*). Stimulation-dependent formation of bulk endosomes and their progressive disappearance during recovery was also similar in the two groups of neurons (*Figure 11F*). In summary, KD of CHC in Dyn1/3 DKO neurons does not enhance the inhibitory effect on SV recovery when bulk endocytosis is the predominant form of endocytosis.

It was proposed that formation of SVs from nerve terminal endosomes could occur via AP-1 and/or AP-3-mediated pathways sensitive to brefeldin A (*Faúndez et al., 1998*; *Glyvuk et al., 2010*; *Cheung and Cousin, 2012*). However, generation of SVs from bulk endosomes during 1 hr recovery still occurred robustly in the presence of brefeldin A (25 µg/ml) (*Figure 12*). Likewise, SV reformation in the presence of latrunculin B (5 µM) argues against an essential role of F-actin (*Figure 12*), although actin may be important for bulk endocytosis.

## Discussion

Our results demonstrate that SV reformation following a strong, acute, secretory stimulus (either high K⁺ depolarization or high frequency electrical activity) occurs predominantly via the budding of SVs from endosomal intermediates generated by bulk endocytosis (bulk endosomes). They show that formation of bulk endosomes robustly occurs in the absence of the two dynamin isoforms (dynamin 1 and 3) that collectively account for the overwhelming majority of neuronal dynamin (*Ferguson et al., 2007*; *Raimondi et al., 2011*). They further strongly suggest that clathrin is not essential for the formation of SVs from bulk endosomes (*Figure 13*).

By extension, these data rule out an essential role for the phosphorylation–dephosphorylation cycle of dynamin 1 in bulk endocytosis (*Anggono et al., 2006*; *Armbruster et al., 2013*), a process that controls the interaction of dynamin 1 with syndapin/pacsin (*Anggono et al., 2006*). This was surprising, as previous studies had proposed a role of the calcineurin-dependent dephosphorylation of dynamin 1 in bulk endocytosis (*Clayton et al., 2009, 2010*; *Xue et al., 2011*). Since dynamin 2 is expressed in Dyn1/3 DKO neurons, dynamin 2 could in principle account for the bulk endocytosis occurring in these cells. However, dynamin 2 represents only a very minor fraction of total brain dynamin and its levels are not enhanced in Dyn1/3 DKO neurons (*Raimondi et al., 2011*). While this issue could be solved by the conditional disruption of all the dynamins in neurons, the generation of dynamin triple KO synapses has so far proven unfeasible due to the very long half-life of dynamin in axon endings and to the neuronal death that occurs before nerve endings are completely depleted of dynamin 2 (our unpublished observations). The complete KO of all three dynamins was achieved in fibroblasts, as these cells can survive for weeks without any dynamin (*Park et al., 2013*). Even in triple KO fibroblasts, other forms of endocytosis were not affected, while CME was blocked (*Ferguson et al., 2009*; *Park et al., 2013*). We suggest that the two neuronal dynamins are specially adapted to support CME during stimulation, and that dynamin 2 may account for CME under resting conditions. This

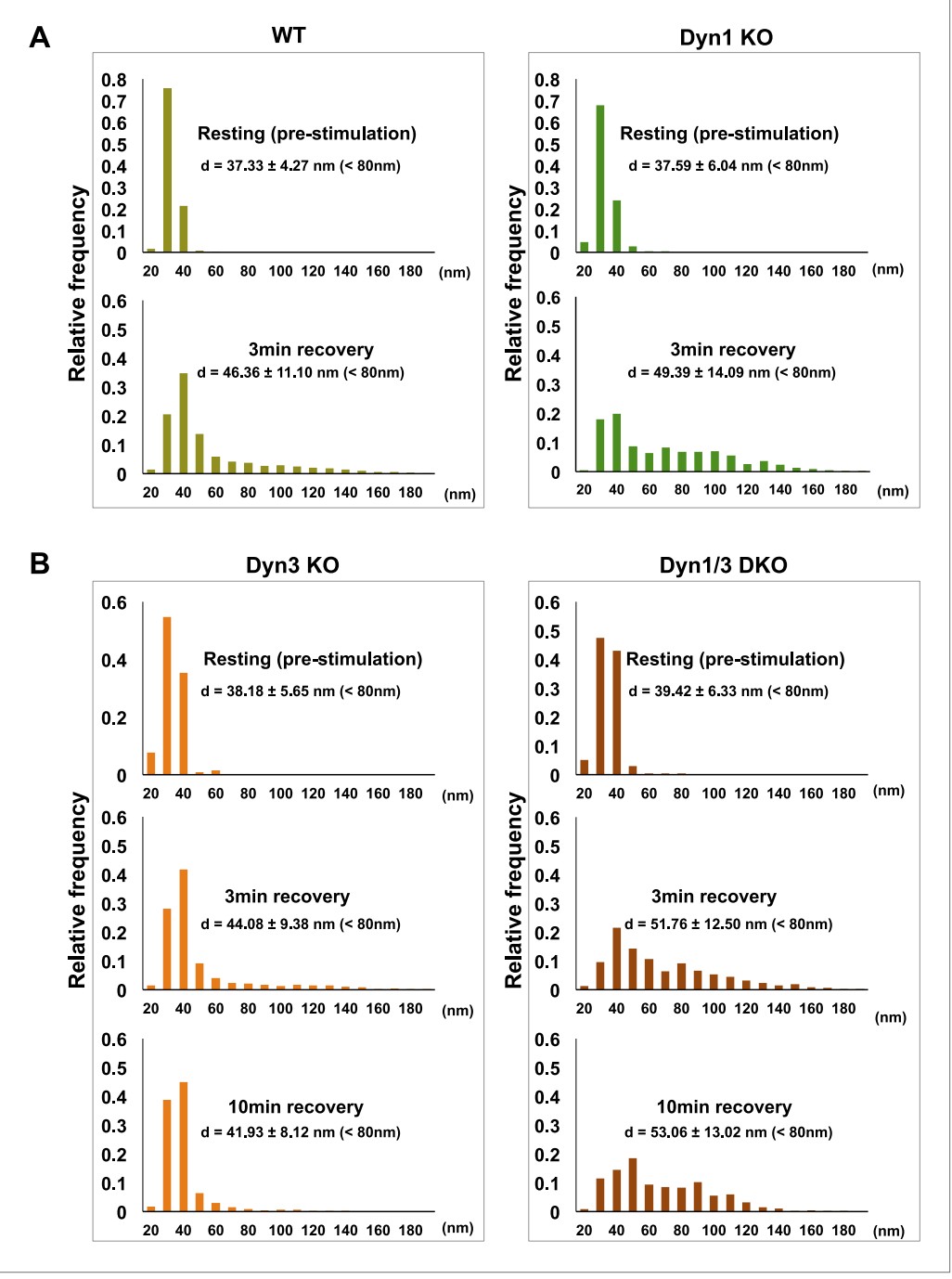

**Figure 10**. SVs reformed after a wave of bulk endocytosis have a larger and more heteregenous average diameter. Neuronal cultures of the indicated genotypes were stimulated with high K⁺ for 90 s and processed by high pressure-freezing/freeze-substitution EM. The diameter of all vesicles (within the 20–200 nm range) in sections of individual nerve terminals before stimulation and after 3 and/or 10 min recovery were analyzed. (**A**) WT and Dyn1 KO nerve terminals. (**B**) Dyn3 KO and Dyn1/3 DKO nerve terminals. Diameters were binned at 10 nm intervals and the average diameters (± SD) of vesicles smaller than 80 nm (i.e., vesicles considered as SVs) are indicated in each field. Note (a) the shift of the peak of SV diameter from the 30 to the 40 nm bin, (b) the larger average SV size and (c) the larger SD during the recovery after the stimulus in all genotypes. Nerve terminals analyzed: 21 for WT R, 106 for WT 3 min, 18 for Dyn1 KO R, 123 for Dyn1 KO 3 min, 20 for Dyn3 KO R, 86 for Dyn3 KO 3 min, 27 for Dyn3 KO 10 min, 17 for Dyn1/3 DKO R, 114 for Dyn1/3 DKO 3 min, 42 for Dyn1/3 DKO 10 min.

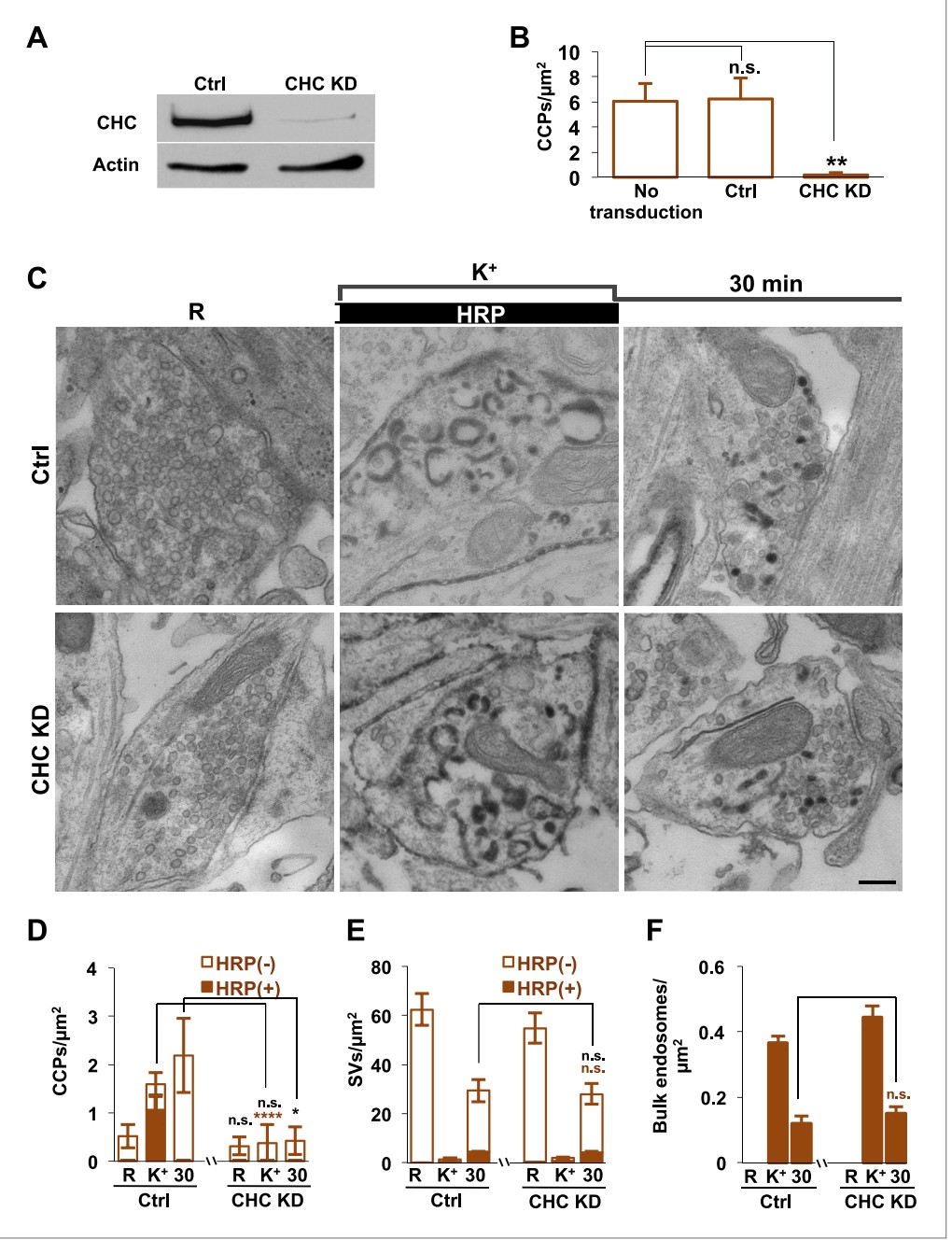

**Figure 11**. Clathrin heavy chain (CHC) depletion does not further impair bulk endosome conversion to SVs in Dyn1/3 DKO neurons. (**A**) Western blotting of extracts of Dyn1/3 DKO neuronal cultures showing depletion of CHC by transduction with lentivirus encoding CHC shRNAmiR but not by lentivirus encoding control shRNAmiR. (**B**) Drastic reduction in CCP number in axon terminals of Dyn1/3 DKO neurons maintained under spontaneous network activity upon CHC KD. (**C–F**) Impact of CHC KD on the ultrastructure of Dyn1/3 DKO neurons preincubated overnight in TTX, stimulated with high K+ for 90 s in the presence of soluble HRP and then allowed to recover for 30 min in HRP-free medium. (**C**) Electron micrographs showing that the KD of CHC does not prevent the reformation of SVs after stimulation in Dyn1/3 DKO neurons. (**D**) High K+ stimulation induces an increase of CCPs number in Dyn1/3 DKO neurons that express control shRNAmiR but not in neurons that express CHC shRNAmiR (CCPs had already been reduced to very low level also in the control treated cultures before stimulation by the overnight treatment in TTX). (**E**) The recovery of SV number is similar in control and CHC KD conditions. (**F**) The reduction of bulk endosomes during recovery at 30 min is not affected by CHC KD. The values of SVs, CCPs and bulk endosomes per μm$^2$ were calculated by averaging the corresponding values calculated at individual synapses.
*Figure 11. Continued on next page*

*Figure 11. Continued*

Scale bar = 200 nm. ****, **, * indicate p-values of <0.0001, <0.01 and <0.05, respectively. n.s., 'not significant'. Black asterisks refer to comparisons between total vesicles, colored asterisks to comparisons between HRP-labeled vesicles. Bars: standard error of the mean.

would explain why the massive accumulation of clathrin coated pits observed in Dyn1/3 KO cultures slowly disappear upon electrical silencing of the cultures.

The occurrence of robust SV membrane endocytosis at Dyn1/3 DKO synapses in spite of the dramatic reduction of dynamin levels also contrasts with the potent global block of the endocytic retrieval of SV membranes in *shibire*[ts] (dynamin) mutant flies at the restrictive temperature (*Kosaka and Ikeda, 1983*; *Koenig and Ikeda, 1989*; *Ramaswami et al., 1994*). One potential explanation is that mutant dynamin produces a potent dominant negative effect by sequestering dynamin interactors. Likewise, the reported strong global blocking effect on SV endocytosis of drugs that impair dynamin action (*Newton et al., 2006*; *McCluskey et al., 2013*) may be explained by the prominent off-target effects of such drugs (*Park et al., 2013*). Our results do suggest the involvement of dynamin 1, and by extension its stimulation-dependent dephosphorylation at synapses, in the reformation of SVs that occurs during stimulation, as this process, but not bulk endocytosis, is strikingly abolished by the absence of dynamin 1. This effect could be explained by a complete block of CME during stimulation, as our studies

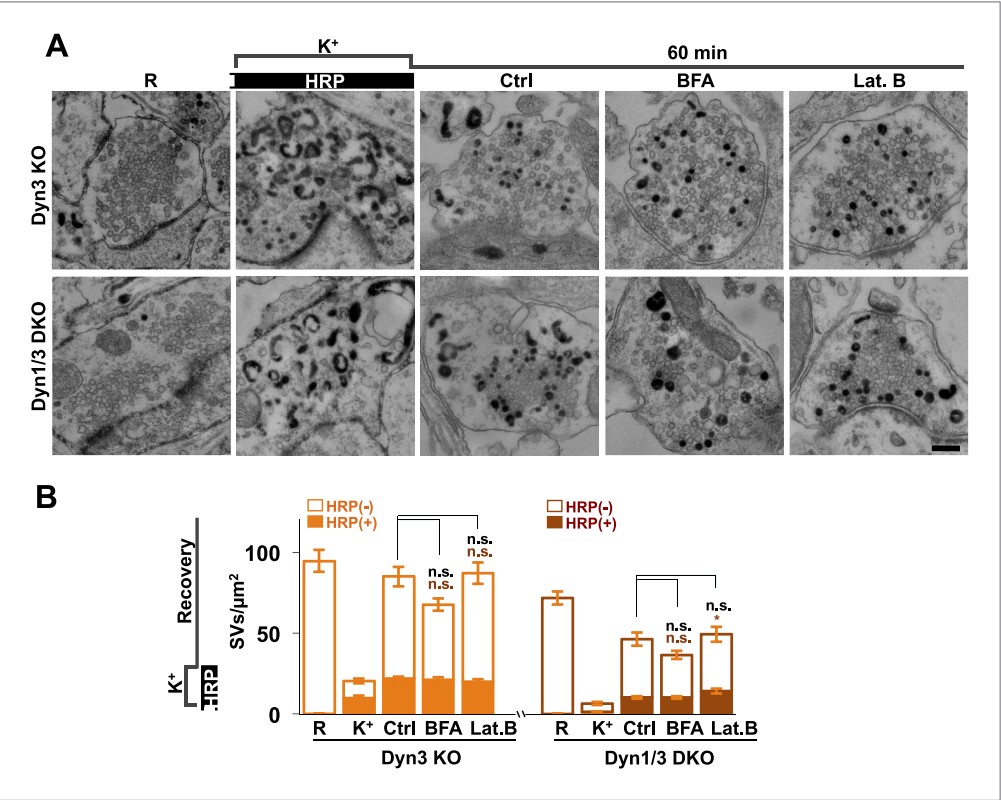

**Figure 12**. No major effect of Brefeldin A and Latrunculin B treatments on SV recovery in Dyn3 KO and Dyn1/3 DKO neurons. Ultrastructure of Dyn1/3 DKO neurons preincubated overnight in TTX, stimulated with high K+ for 90 s in the presence of soluble HRP and then allowed to recover for 60 min in HRP-free medium. (**A**) Representative EM micrographs and (**B**) quantification of the results showing that recovery of SVs after 1 hr is similar in controls and in the presence of BFA (25 µg/ml) or Lat. B (5 µM), though a modest statistical significance was found for total SVs recovery in BFA-treated neurons and HRP-labeled SVs in Lat.B treated conditions. SVs positive for HRP reaction product are indicated by the dark portion of each bar. Scale bar = 200 nm. *p<0.05, n.s., 'not significant'. Black asterisks refer to comparisons between total vesicles, colored asterisks to comparisons between HRP-labeled vesicles. Bars: standard error of the mean.

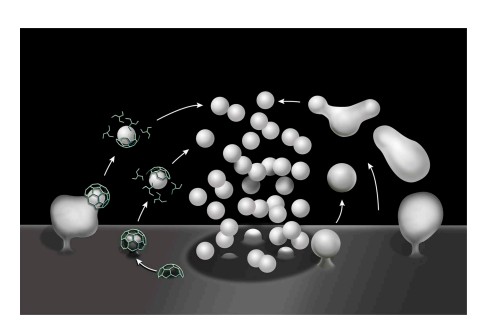

**Figure 13**. Schematic drawing illustrating clathrin-mediated and clathrin independent endocytosis of synaptic vesicle membranes. At left is CME. At right is bulk endocytosis followed by fragmentation of bulk endosomes to generate new vesicles. As membranes of bulk endosomes are expected to be enriched in SV proteins, functional SVs may be generated by this mechanism. The uncoated invaginations occurring near active zones and described by Watanabe et al. (*2013a* and *2013b*) may represent a form of this type of clathrin-independent endocytosis. At subsequent exo–endocytic cycles, however, CME may help reestablish SV reformation with high fidelitiy.

of dynamin mutant cells have consistently supported a critical role of dynamin in CME (*Ferguson et al., 2007*, *2009*; *Hayashi et al., 2008*; *Raimondi et al., 2011*). Since a block of SV endocytosis, as detected by a synaptopHluorin assay (*Sankaranarayanan and Ryan, 2001*) was observed at Dyn1 KO and Dyn1/3 DKO synapses during moderate frequency electrical stimulation (300 stimuli at 10 Hz), bulk endocytosis was likely not triggered by those experimental conditions.

Concerning clathrin, its essential role in SV reformation (in pathways other than kiss-and-run) was challenged by genetic studies in *C. elegans*, as synaptic transmission was not abolished in clathrin mutant worms (*Sato et al., 2009*). Furthermore, recent studies of *C. elegans* (*Kittelmann et al., 2013*; *Watanabe et al., 2013a*) and mouse synapses (*Watanabe et al., 2013b*) expressing channelrhodopsin and subjected to photostimulation revealed clathrin-independent forms of endocytosis mediated by vacuolar invaginations larger than SVs (*Figure 13*). The endocytic reaction described by *Kittelmann et al. (2013)* following strong photostimulation is reminiscent of the bulk endocytosis described here. Interestingly, in that study the disappearance of the large vacuoles produced by the intense stimulus was delayed in

a dynamin$^{ts}$ mutant at the restrictive temperature, reminiscent of the delay in the conversion of bulk endosome to SVs observed here at Dyn1/3 DKO synapses. In contrast, the ultrafast endocytic reaction reported by Watanabe et al. (*2013a*, *2013b*) in either *C. elegans* or mouse neurons following a very short photostimulus was reported to be dependent on dynamin, at variance with the endocytic reaction observed in our study.

The mechanisms through which membranes of bulk endosomes are converted to SVs remain elusive. SVs are a highly specialized form of small vesicular carriers present in all cells. The prevailing general mechanism for vesicle budding in the secretory and endocytic pathways is the assembly on the donor membrane compartments of a 'coat', whose core and accessory subunits are responsible for two major functions: induction of bilayer deformation and protein cargo selection (*Faini et al., 2013*). Among the well characterized coats, the only one so far for which there is evidence in SV generation from genetic, biochemical and microscopic observations, is the clathrin coat (*Heuser and Reese 1973*; *Maycox et al., 1992*; *Shupliakov et al., 1997*; *Nonet et al., 1999*; *Verstreken et al., 2002*; *Heerssen et al., 2008*). However, our study suggests that clathrin-mediated buds are present only, or at least by far predominantly, on the nerve terminal plasma membrane. Furthermore, SV reformation from bulk endosomes occurs at Dyn1/3 DKO synapses is not further delayed after CHC KD.

Several membrane-associated scaffolds that mediate budding at the surface of endosomes have been described in recent years (*Bonifacino and Rojas, 2006*; *Schellmann and Pimpl, 2009*), but none of them, so far, has been shown to have a critical function in synaptic transmission. The membranes of bulk endosomes are expected to be highly enriched in SV proteins, as they form by compensatory endocytosis after SV fusion. Thus, protein cargo selection may not be a critical function in this process and mechanisms that force membrane curvature may be sufficient to induce SV formation, albeit perhaps leading to SVs with a non-optimal protein composition and size. For example, proteins containing BAR domains (*Frost et al., 2009*; *Campelo et al., 2010*) or other membrane deforming proteins for which there is evidence in SV recycling may be involved. Direct reformation of SVs from beading and 'imprecise' cutting of tubular structures is a possibility, as supported by the slightly larger and more heterogeneous size of the SVs upon recovery from a massive stimulation. Peripheral membrane proteins such as the synapsins, which are enriched at the surface of SVs (*De Camilli et al., 1983*), but are shed during the endocytic phase of their recycling, may also help budding of SVs from bulk

endosomes via their bilayer penetrating ALPS motifs (*Krabben et al., 2011*). Lipid-mediated mechanisms, that is metabolic changes in the bilayer leading to its spontaneous acquisition of curvature should also be considered. Importantly, however, we suggest that this mode of SV recycling functions only as a back-up mechanism during intense activity. At subsequent exo-endocytic cycles, CME may help regenerate SVs with precise size and molecular composition.

An important open question is why the conversion of bulk endosomes into SVs is delayed in mammalian synapses that lack dynamin 1 and 3 (our study) and in dynamin mutant worms (*Kittelmann et al., 2013*). A direct role for dynamin in this process cannot be excluded, since in yeast the dynamin homologue Vps1 was implicated in endosomal sorting (*Ekena et al., 1993*; *Chi et al., 2014*) and some studies have suggested roles for dynamin on endosomes in mammalian cells (*Hayden et al., 2013*). However, the conversion of bulk endosomes into SVs still occurs, albeit less efficiently, even in the absence of the two major neuronal dynamin isoforms, pointing to potential dynamin independent mechanisms. It is also possible that the delay in the conversion from bulk endosomes to SVs observed in Dyn1/3 DKO synapses may be an indirect effect of impaired dynamin function. For example, the overload of this pathway in the absence of CME may saturate it. Alternatively, the absence of dynamin may produce changes in signaling pathways that regulate SV formation from bulk endosomes. This possibility is supported by the changes that we previously observed in synapsin phosphorylation in Dyn1/3 DKO synapse (*Raimondi et al., 2011*), which may in turn affect bulk endosome dynamics. Elucidating the mechanisms underlying the conversion of bulk endosomes into SVs will not only advance the field of synaptic transmission but also provide key new insight in fundamental mechanisms in vesicle biogenesis.

## Addendum

While the revision of this manuscript was being finalized, a paper by *Kononenko et al. (2014)* reported that bulk endocytosis in response to a strong stimulus is dynamin 1/3 dependent (based on RNA interference) and proposed an important role for clathrin in SV budding from endosomes. Our results, which are based on studies of dynamin 1/3 double KO synapses and on the ultrastructrual analysis of the time course of SV recovery from bulk endosomes in the presence or near depletion of clathrin lead to different conclusions.

# Materials and methods

## Gene targeting strategies

The generation of *DNM1−/−*, *DNM3−/−*, and *DNM1−/−; 3−/−* mice was described previously (*Ferguson et al., 2007*; *Raimondi et al., 2011*).

## Neuronal cultures and their stimulation

Primary cultures of cortical neurons were prepared from E18 day mouse brains as described (*Ferguson et al., 2007*) and analyzed after 2–3 weeks in vitro. Neurons were maintained in Neurobasal medium supplemented with B27 (2%), Glutamax (2 mM) and penicillin/streptomycin at 37°C (all media components were from Gibco, Life Technologies, Grand Island, NY). Prior to stimulation and labeling experiments, cultures were incubated overnight with 1 μM TTX to silence synaptic activity and thus to reverse the accumulation of endocytic CCPs which is characteristic of neurons defective in dynamin function (see *Ferguson et al., 2007*; *Hayashi et al., 2008*; *Raimondi et al., 2011*). After a rinse with Tyrode buffer without $Ca^{2+}$ (119 mM NaCl, 2.5 mM KCl, 2 mM $MgCl_2$, 25 mM Hepes, and 30 mM glucose, pH 7.4) to wash out TTX, cultures were pre-incubated for 5 min with HRP (10 mg/ml, Sigma-Aldrich, St. Louis, MO) or CTX-HRP (10 μg/ml, Molecular Probes, Life Technologies, Eugene, Oregon) in the same buffer. Subsequently, they were stimulated by incubation for 90 s, this time with 2 mM $Ca^{2+}$, but with a high concentration of $K^+$ (90 mM) and a corresponding reduction in $Na^+$. Alternatively, after a rinse with Tyrode buffer, coverslips were placed into a Chamlide stimulation chamber (Live Cell Instrument, Seoul, South Korea) and neuronal cultures were subjected to 10 s electrical field stimulation at 80 Hz (100 mAmp, 1 ms pulse) using a Master-8 pulse stimulator (AMPI, Jerusalem, Israel). Cultures were then washed rapidly three times in $Ca^{2+}$-free Tyrode buffer and subsequently allowed to recover under conditions that prevent synaptic activity, either $Ca^{2+}$ free Tyrode buffer also containing 0.5 mM EGTA or Tyrode buffer containing 1 μM TTX. HRP or CTX-HRP were present during the stimulus, the recovery period of both, as described in the text. All procedures involving mice were approved by the Yale University Institutional Animal Care and Use Committee.

## Electron microscopy and electron tomography

Cells were chemically fixed with 1.2% glutaraldehyde in 0.1 M sodium cacodylate buffer, postfixed with 1% $OsO_4$, 1.5% $K_4Fe(CN)_6$, (Sigma-Aldrich, St. Louis, MO), in the same buffer, *en bloc* stained with 0.5% uranyl magnesium acetate, dehydrated, and embedded in Embed 812 (all EM experiment components are from EMS, Hatfield, PA). The HRP reaction was developed with diaminobenzidene and $H_2O_2$ after the glutaraldehyde fixation step. Ultrathin sections were observed in a Philips CM10 microscope at 80 kV and images were taken with a Morada 1 k × 1 k CCD camera (Olympus). Synapses were selected based on the presence of an active zone. SV number and the membrane area of HRP-labeled bulk endosomes within the 60 nm thick section (endosome perimeter multiplied by 60) were measured from 30 to 100 synapses for each condition (1–3 pups for each condition depending on the genotypes found in the litter) and experiments. These values were normalized to the cross-sectional area of the presynaptic terminal. Results of the morphometric analysis are presented as average ± SEM, with the exceptions of data shown in *Figure 10* which are shown as average ± SD. Significance for *Figures 2, 4, 6 and 11* was evaluated using two-way ANOVA followed by Sidak's multiple comparisons test. Significance for *Figures 1, 5 and 11B, 12* was evaluated with one-way ANOVA followed by Dunnett's or Tukey multiple comparisons test.

Electron tomography was performed as described (*Hayashi et al., 2008*) on 200–250 nm thick sections using either a TECNAI F20 or a F30 intermediate-voltage microscope operating at 200 kV or 300 kV (Boulder Laboratory for 3-D Electron Microscopy of Cells, University of Colorado). For the generation of the final models, contours of membranes were traced manually, but small vesicular structures SVs, CCVs, and CCPs were displayed as spheres.

## High-pressure freezing and freeze substitution

Neurons were cultured on the opposite side of 20 nm carbon-coated 3 mm sapphire discs (TECHNOTRAD, Manchester, NH) (scratched with a unsymmetric symbol to distinguish sidedness). For freezing, discs were removed from the culture dishes, sandwiched between the 100 μm side of an A hat and the flat side of a B hat (TED PELLA, Red, CA) and flash frozen under high pressure with a Leica EM HPM100 machine with or without 20% BSA (Sigma-Aldrich, St. Louis, MO). All these manipulations were performed within 20–30 s. Frozen neurons were transferred to liquid nitrogen for storage. For freeze substitution (*Watanabe et al., 2013b*), samples were transferred to a Leica EM AFS2 apparatus, where they were first immersed in acetone containing 1% OsO4, 1% glutaraldehyde and 1% $H_2O$ at −90°C for 1-3 hr. Subsequently, they were slowly shifted (with temperature increases of 10°C/hr) to −20°C, kept for 12 hr at this temperature, and then further shifted to +20°C (again with temperature increases of 10°C/hr). Samples were then washed in acetone containing 1% $H_2O$ (10 min × 3), incubated with 0.1% uranyl acetate +1% $H_2O$ in acetone for 0.5–1 hr at room temperature (RT), washed in pure acetone at RT (10 min × 3) and infiltrated at RT with epon using the following sequence: 1 hr epon/acetone 1:2 (vol/vol); 1 hr epon/acetone 2:1 (vol/vol); 2 hr pure epon at RT. Finally, embedded samples were polymerized at 60°C for 48 hr.

## Lentivirus construction

For CHC KD, a target sequence for mouse CHC (5′-GAAGATAAGCTGGAATGTTCT) was designed using the Block-iT RNAi Designer (Invitrogen, Life Technologies, Canada) and subcloned into pcDNA6.2/GW-EmGFP-miR (Invitrogen, Life Technologies, Canada) following manufacturer's instructions. The EmGFP-miR cassette was amplified by PCR and subcloned into the pRRLsinPPT vector to generate the lentiviral expression vector. The control virus has been described previously (*Thomas et al., 2009*). VSV-G pseudo-typed lentiviral particles were prepared as described previously (*Ritter et al., 2013*).

## Acknowledgements

We thank Ruben Fernandez-Busnadiego, Andrea Raimondi, Abel Alcazar and Summer Paradise for discussion and help, Frank Wilson, Louise Lucast and Lijuan Liu for outstanding technical assistance, and Morven Graham (Yale Center for Cell and Molecular Imaging) for assistance with EM. This work was supported in part by grants from the NIH (NS36251, DK45735, and DA018343 to PDC and P41GM103431 to A Honger), from the Howard Hughes Medical Institute (HHMI), the Brain and Behavior Research Foundation (formerly NARSAD) and the Ellison Medical Foundation to PDC and from the Canadian Institute of Health Research (MOP-13461) to PSM.

# Additional information

## Funding

| Funder | Grant reference number | Author |
|---|---|---|
| National Institutes of Health | NS36251, DK45735, DA018343 | Pietro De Camilli |
| Canadian Institutes of Health Research | CIHR MOP-13461 | Peter S McPherson |
| Brain and Behavior Research Foundation | | Pietro De Camilli |
| Howard Hughes Medical Institute | | Pietro De Camilli |
| Ellison Medical Foundation | | Pietro De Camilli |
| National Institutes of Health | P41GM103431 | Eileen T O'Toole |

The funders had no role in study design, data collection and interpretation, or the decision to submit the work for publication.

## Author contributions

YW, Conception and design, Acquisition of data, Analysis and interpretation of data, Drafting or revising the article; ETO'T, XL, Helped with electron tomography, Acquisition of data; MG, BR, Generated and validated viral vectors, Contributed unpublished essential data or reagents; MM, Electrical stimulation experiments, Acquisition of data; PSM, Drafting or revising the article, Contributed unpublished essential data or reagents; SMF, PDC, Conception and design, Analysis and interpretation of data, Drafting or revising the article

## Ethics

Animal experimentation: The present study was performed in strict accordance with the recommendations in the Guide for the Care and Use of Laboratory Animals of the National Institutes of Health. All of the animals were handled according to approved institutional animal care and use committee (IACUC) protocols (#07422-2012) of Yale university. All animals were euthanized with $CO_2$ and embryos were used for primary neuron culture.

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
