## [Decision Letter]

Thank you for sending your work entitled “A clathrin and dynamin-independent pathway of synaptic vesicle recycling mediated by bulk endocytosis” for consideration at *eLife.* Your article has been favorably evaluated by a Senior editor and 3 reviewers, one of whom is a member of our Board of Reviewing Editors.

The Reviewing editor and the other reviewers discussed their comments before we reached this decision, and the Reviewing editor has assembled the following comments to help you prepare a revised submission.

In this work, De Camilli and coworkers take advantage of the mice deficient in dynamin 1 and dynamin 3 to address an old and still controversially discussed problem of synaptic vesicle recycling, namely the role of bulk endocytosis (observed as early as in the seventies) in synaptic vesicle recycling and its relationship to the clathrin- and dynamin-dependent pathway. The authors use cultured neurons that are subjected to a massive, vesicle-depleting stimulus and then allowed to recover exocytosed membranes by endocytosis, partially in the presence of extracellular tracers. Analysis is carried out by quantitative EM, with conventional fixation/staining protocols being complemented by cryo-EM following high-pressure freezing.

All referees agreed that the manuscript makes an interesting contribution to a longstanding and controversial debate. However, several of us had substantial concerns, and while in the end we decided to allow for revision, the following points should be addressed:

1) Stimulation by KCl is highly non-physiological and thus the intermediate structures described here may only become apparent at extreme stimulation conditions. We therefore suggest repeating at least some of the key experiments using stimulation by action potentials (at least for Figure 1).

2) There was considerable debate among the referees whether the remaining proteins (dynamin 2, residual clathrin) may not be sufficient to maintain some function despite very low protein levels. There are precedents for this: e.g. in MEFs, Parkin is only detectable by Q-RT-PCR and it is undetectable by western, yet this very low amount of Parkin is enough to support mitophagy (in parkin null MEFs mitophagy is blocked). While we realize that this issue is not easy to deal with, there should be at least an open discussion of this possibility. Also, the authors may try to carry out knockdown of dynamin 2 for further reduction of dynamin levels.

3) While the authors state that synaptic vesicles are being re-formed from the endosomal intermediates, the morphological distinction between endosomes and synaptic vesicles is not clear (most of them appear larger – were the diameters quantified, and where is the cutoff?). Also, evidence should be provided that these regenerated synaptic vesicles are capable of undergoing exocytosis.

In the following, we include some of the original referee comments to explain the concerns listed above in more detail:

Reviewer #1:

1) (...) The conclusion that synaptic vesicles are able to recycle without clathrin or without dynamin largely stems from the observation that bulk endosomes resolve into small synaptic vesicles in dynamin 1/3 double knock outs or upon additional clathrin knock down. However, not all dynamin function is absent and dynamin 2 is still present in these experiments. The authors argue that dynamin2 is present at low levels, but is very difficult to infer function simply from protein levels. Maybe a couple of percent of dynamin 2 are enough to support the very slow vesicle recycling the authors are observing. This point is further substantiated by the fact that triple knock out neurons do not survive while dynamin 1/3 double knock outs do, indicating an important function for the dynamin 2 protein despite the fact that it is present at only very low levels. A similar argument holds true for clathrin knock down: some clathrin is still persisting; there are even a few clathrin coated pits still observed after knock down, underscoring the presence of some clathrin protein upon knock down (Figure 8). I understand that it is at present not possible to generate neurons that lack dynamin (or clathrin) completely, but because of this, the interpretation that synaptic vesicles can be formed in the absence of either protein lacks experimental support and the alternative interpretation that slow 'classical' endocytosis mediates the recycling of the synaptic vesicles seems possible and even more realistic.

2) I don't follow the argumentation that in dynamin1/3 KOs many of the synaptic like vesicles that form during the recovery period are not HRP labeled? The authors argue that stochastic labeling of all-or-non DAB conversion could explain this observation. In controls the same issues would be at play, yet there, almost all synaptic vesicles are labeled?

3) The HPF EM data shows bulk endosomes that sometimes display tubular extensions. Based on these morphological features the authors argue that these extensions my yield synaptic vesicles. The authors observe a dense coat at the tips of these extensions and claim this is not clathrin. This qualitative statement is not convincing and additional evidence that this is or is not clathrin is needed to make a convincing argument. In addition, it seems that the diameter of these structures does not match the diameter of an average synaptic vesicle.

Reviewer #2:

The authors suggest that synaptic vesicles form after stimulation of DKO synapses. In their analysis figures (for example Figure 2), the authors present separate graphs for bulk endosomes and synaptic vesicles. It is not clear how the authors decided whether a specific organelle is a synaptic vesicle or a bulk endosome. I used some of the PDF images provided in the submission, and analyzed the diameters of labeled organelles in DKO and WT synapses. I performed this for two figures in which numerous such organelles are included, Figure 1 and Figure 3. I only analyzed the 30 minute recovery images. In Figure 1 found that 57 of 60 organelles in the WT synapse had diameters of 10 pixels or below, corresponding to synaptic vesicles. In contrast, 18 of the 19 organelles I analyzed in the DKO synapse from Figure 1 had diameters of more than 10 pixels. I would interpret this as an indication that these synapses do not form synaptic vesicles, but rather smaller endosomes. This discrepancy is not as large in Figure 3, where about one third of the labeled DKO organelles are within the size range of the labeled WT organelles. Nevertheless, one can still conclude that the organelles forming in the DKO synapses are typically larger than synaptic vesicles. The authors should therefore include a characterization of the sizes of the vesicles forming during recovery after stimulation, and should indicate the amount of such vesicles that are within the size range of normal, WT synaptic vesicles.

Based on these measurements, a clear cutoff should be defined between synaptic vesicles and bulk endosomes (to be used in analysis figures such as Figure 2).

Along the same lines, the authors need to determine whether the small vesicles forming in DKO synapses exhibit the physiological characteristics of synaptic vesicles. For example, can these vesicles exocytose upon stimulation? If not, then this pathway cannot be suggested to form true synaptic vesicles. A number of strategies could be tried here, including pHluorin experiments and FM dye uptake and release.

---

## [Author Response]

As we were about to finalize our material for resubmission, a paper on a similar topic appeared in Neuron: Clathrin/AP-2 Mediate Synaptic Vesicle Reformation from Endosome-like Vacuoles but Are Not Essential for Membrane Retrieval at Central Synapses. 2014. Kononenk, Puchkov, Classen,Walter, Pechstein, Sawade, Kaempf, Trimbuch, Lorenz, Rosenmund, Maritzen and Haucke. Neuron 82, 981–988.

Their major conclusions are quite contrary to what our experiments have revealed. While they show, like us, that high frequency stimulation triggers bulk endocytosis, they conclude, based on RNA interference, that bulk endocytosis requires both dynamin 1 and 3 and that reformation of synaptic vesicles from bulk endosomes is a clathrin-dependent process. We note the following:

1) Our data proves beyond any reasonable doubt (we use double KO mice) that dynamin 1 and 3 are not needed for bulk endocytosis; this process is actually enhanced in the complete absence (germline KO) of these two neuronal dynamins, possibly because clathrin-mediated endocytosis is impaired. In contrast, Kononenk et al based their conclusions about the requirement for dynamin 1 and 3 in bulk endocytosis on RNA interference experiments that lacked even minimal controls to establish that the observed phenotypes were due to on target effects of the shRNA. Furthermore, their conclusions are indirect, based on pHluorin experiments, and not supported by ultrastructural studies.

2) While the manuscript by Kononenk et al concludes that clathrin is required for the conversion of bulk endosomes to SVs, it does not provide direct evidence for the presence of clathrin on bulk endosomes. They show that after clathrin knockdown, stimulation is followed by the accumulation of endosome-like vacuoles (bulk endosomes). In contrast to their conclusion that clathrin must be required for the conversion of bulk endosomes to SVs, this observation is equally consistent with the possibility that bulk endosomes form more extensively when clathrin mediated budding of SVs from the plasma membrane is impaired.

As this other paper just appeared (June 4), we did not include it in our review of “prior knowledge”. We have made a reference to it in an Addendum at the end of our manuscript:

Addendum: While the revision of this manuscript was being finalized, a paper by [40] reported that bulk endocytosis in response to a strong stimulus is dynamin 1/3 dependent (based on RNA interference and pHluorin based experiments) and proposed an important role for clathrin in SV budding from endosomes. Our results, which are based on ultrastructural studies of dynamin 1/3 double KO synapses and on the ultrastructrual analysis of the time course of SV recovery from bulk endosomes in the presence or near depletion of clathrin, lead to different conclusions.

Response to the comments of the reviewers:

We found the review to be very constructive and we have now addressed all the comments with new experimentation. The delay in responding relative to the time when we received the decision letter is explained by the long-term nature of the work involved in this research. Experiments rely on i) the breeding of mice that must be kept at the heterozygous state for the dynamin 1 allele, ii) 2-3 weeks for each preparation of neuronal primary cultures to mature and iii) a time consuming analysis by EM (morphometry and reconstruction of EM tomograms).

New data include: 1) evidence the SVs reformed by bulk endocytosis are available for a new round of release, thus indicating that they are *bona fide* functional synaptic vesicles; 2) quantitative analysis of endocytic intermediates following action potential stimulation in addition to high K^+^ stimulation; 3) new high pressure freezing experiments, providing improved morphological preservation (with corresponding new tomographic analysis); 4) analysis of synaptic vesicle size. In addition 5) statistical data have been added to all the figures reporting morphometric data. Figure 5, Figure 6, Figure 9, Figure 10 and Figure 9—figure supplement 1 to Figure 9 are completely new. We have also added a final cartoon to summarize our findings (new Figure 13).

Two new authors have been added, Mirko Messa and Xinran Liu, as they contributed to the new electrical stimulation and tomographic experiments.

Points that needed to be addressed based on the editorial letter.

*1) Stimulation by KCl is highly non-physiological and thus the intermediate structures described here may only become apparent at extreme stimulation conditions. We therefore suggest repeating at least some of the key experiments using stimulation by action potentials (at least for*
Figure 1*)*.

We performed these experiments. We stimulated dyn3KO (CTRL) and dyn1/3DKO cultures at 80 Hz for 10 s in the presence of CTX-HRP. EM analysis revealed that the bulk endocytosis and recovery occurring under these conditions had similar properties to that observed with high K^+^ stimulation. These data are now shown in revised Figure 5

*2) There was considerable debate among the referees whether the remaining proteins (dynamin 2, residual clathrin) may not be sufficient to maintain some function despite very low protein levels. There are precedents for this: e.g. in MEFs, Parkin is only detectable by Q-RT-PCR and it is undetectable by western, yet this very low amount of Parkin is enough to support mitophagy (in parkin null MEFs mitophagy is blocked). While we realize that this issue is not easy to deal with, there should be at least an open discussion of this possibility. Also, the authors may try to carry out knockdown of dynamin 2 for further reduction of dynamin levels*.

We invested great effort (even before submitting this manuscript) toward the generation or “null” dynamin synapses. We have in our lab the conditional KO mice of the appropriate genotype needed to achieve this goal and we used such mice to generate conditional triple KO fibroblasts (reported in Park et al. “Dynamin triple knockout cells reveal off target effects of commonly used dynamin inhibitors”. J Cell Sci. 2013:5305-12). However, we found that while mouse fibroblasts survive for weeks, even after all dynamin alleles have been disrupted and all dynamin was lost, neurons die very rapidly when all dynamin alleles are disrupted. For example, expression of Cre (either via the Cre-ER-tamoxifen methodology or via viral transduction) in cultured dynamin 1/ dynamin 3 double KO neurons harboring floxed dynamin 2 alleles resulted in neuronal death within few days, before all dynamin 2 could disappear from nerve terminals (Given the distance of nerve terminals from cell bodies and dendrites, loss of cytosolic proteins occurs with some delay relative to the loss of the same proteins in perikarya and dendrites). In principle, we could have used pharmacological inhibition to block residual dynamin, as several “anti-dynamin” drugs, primarily dynasore or dyngo, have been reported in the literature.

However, our studies of triple KO fibroblasts demonstrated major off-target effects of these drugs, thus making the use of these drugs unreliable. Thus, we could not answer the question raised here. Similar considerations apply to clathrin as we observed neuronal death before clathrin could completely disappear. Thus, we agree with reviewer 1 that we cannot exclude that slow 'classical' clathrin-mediated endocytosis from the plasma membrane, mediated by dynamin 2, could play a role in mediating the recycling of the synaptic vesicles during recovery from stimulation in the absence of dynamin 1 and 3*.* To comply with these comments, we have now better addressed this issue in the Discussion.

We have also changed the title to “A dynamin 1-, dynamin 3- and clathrin-independent pathway of synaptic vesicle recycling mediated by bulk endocytosis”.

(The original title was “A clathrin and dynamin-independent pathway of synaptic vesicles recycling mediated by bulk endocytosis”.)

*3) While the authors state that synaptic vesicles are being re-formed from the endosomal intermediates, the morphological distinction between endosomes and synaptic vesicles is not clear (most of them appear larger – were the diameters quantified, and where is the cutoff?). Also, evidence should be provided that these regenerated synaptic vesicles are capable of undergoing exocytosis*.

This point deals with the more general issue of what should be defined as an endosome in nerve terminals. The word “endosome” was first introduced by Helenius et al in 1983 (“Endosomes”. Helenius, Mellman, Wall and Hubbard, Trends Bioch, Sci. July 1983: 245-250) to define “a heterogeneous population of endocytic vacuoles through which molecules internalized during pinocytosis pass en route to lysosomes”, and in which molecular sorting occurs. The morphological, biochemical and functional heterogeneity of endosomes has become even more apparent during the last 30 years. In our study we have frequently used the terms “endosomal intermediates” or “bulk endosomes” to emphasize the unique nature of the nerve terminal compartments that form and then disappear in response to massive stimulation. Here we have used size to distinguish such intermediates from synaptic vesicles. We recognize that needs to be precisely defined in the text. We have now defined endosomes as organelles bigger than 80 nm. We have also carried out a new morphometric analysis of vesicular profiles with diameters (in the plane of the section) comprised between 20 and 200 nm at WT and mutant synapses at rest and during recovery. This analysis indicated a slightly larger and more heterogenous diameter of synaptic vesicles in dyn1 KO and dyn1/3 DKO synapses.

We also now provide evidence that SVs regenerated from bulk endosomes are functional by showing that SVs labeled by HRP during a 90 sec high K+ stimulation can undergo exocytosis after a 30 min recovery period upon a second round of high K^+^ stimulation. Newly formed (as defined by HRP labeling) SVs are releasable from both CTRL and dyn1/3DKO neurons.

Although we understand that the points outlined in the editorial letter (above) were the critical ones, we have also addressed to our best additional specific points raised by the reviewers

Reviewer #1:

*1) (...) The conclusion that synaptic vesicles are able to recycle without clathrin or without dynamin largely stems from the observation that bulk endosomes resolve into small synaptic vesicles in dynamin 1/3 double knock outs or upon additional clathrin knock down. However, not all dynamin function is absent and dynamin 2 is still present in these experiments. The authors argue that dynamin2 is present at low levels, but is very difficult to infer function simply from protein levels. Maybe a couple of percent of dynamin 2 are enough to support the very slow vesicle recycling the authors are observing. This point is further substantiated by the fact that triple knock out neurons do not survive while dynamin 1/3 double knock outs do, indicating an important function for the dynamin 2 protein despite the fact that it is present at only very low levels. A similar argument holds true for clathrin knock down: some clathrin is still persisting; there are even a few clathrin coated pits still observed after knock down, underscoring the presence of some clathrin protein upon knock down (*Figure 8*). I understand that it is at present not possible to generate neurons that lack dynamin (or clathrin) completely, but because of this, the interpretation that synaptic vesicles can be formed in the absence of either protein lacks experimental support and the alternative interpretation that slow 'classical' endocytosis mediates the recycling of the synaptic vesicles seems possible and even more realistic*.

We have addressed this question within point 2 of the compiled editorial comments (see above). We agree with this reviewer that residual dynamin, dynamin 2, could play a role in the recycling occurring in the absence of dynamin 1 and 3. In fact, we do believe that it may play some role in clathrin-mediated endocytosis from the plasma membrane still occurring in these cells. However, our data argue against a clathrin/dynamin-dependent classical mechanism from bulk endosomes in dyn1/3DKO neurons, as we do not see clathrin coated pit accumulation on bulk endosomes at synapses of these neurons, in contrast to the robust accumulation of CCPs at the plasma membrane. Tomographic analysis demonstrated that CCPs localized deep into axon endings were in fact localized on deep plasma membrane invaginations. In any case, we have now changed the following sentence of the Abstract by replacing “showing” with “suggesting”: “Conversion of bulk endosomes into SVs during recovery was delayed, but not blocked, even after further clathrin knockdown, showing suggesting that this process is independent of clathrin-mediated budding. We have modified the title and we have been very cautious throughout the text. For example the last sentence of the Introduction (which we have not changed from the original submission) states: “These observations provide evidence for a pathway of SV reformation that does not require dynamin/clathrin mediated endocytosis and that predominates after strong stimulation”.

2) I don't follow the argumentation that in dynamin1/3 KOs many of the synaptic like vesicles that form during the recovery period are not HRP labeled? The authors argue that stochastic labeling of all-or-non DAB conversion could explain this observation. In controls the same issues would be at play, yet there, almost all synaptic vesicles are labeled?

There is some heterogeneity in the fraction of labeled SVs, but, as the quantification data show (Figure 2), such fraction is approximately similar in controls and dynamin 1 KO and dynamin 1/3 DKO synapses. We realize that the original 30 min WT field of Figure 1 could have been misleading (we had chosen samples where most vesicles were labeled to emphasize the occurrence of recycling) and we replaced them.

*3) The HPF EM data shows bulk endosomes that sometimes display tubular extensions. Based on these morphological features the authors argue that these extensions my yield synaptic vesicles. The authors observe a dense coat at the tips of these extensions and claim this is not clathrin. This qualitative statement is not convincing and additional evidence that this is or is not clathrin is needed to make a convincing argument. In addition, it seems that the diameter of these structures does not match the diameter of an average synaptic vesicle*.

We thank the reviewer for this comment. Those tubular structures were indeed very rare and could not be observed in new samples that we have generated with improved morphology. All high pressure-freezing/freeze-substitution images presented are new. The only coated buds observed in vacuolar structures had the characteristic texture of clathrin-coated pits, but these were very few (numbers are now given in results at the description of Figure 9). Furthermore, when tomographic analysis was performed and when the entire ”vacuolar” structure was contained within the tomogram such vacuoles were in fact found to be still connected to the cell surface.

Reviewer #2:

*The authors suggest that synaptic vesicles form after stimulation of DKO synapses. In their analysis figures (for example*
Figure 2*), the authors present separate graphs for bulk endosomes and synaptic vesicles. It is not clear how the authors decided whether a specific organelle is a synaptic vesicle or a bulk endosome. (…) The authors should therefore include a characterization of the sizes of the vesicles forming during recovery after stimulation, and should indicate the amount of such vesicles that are within the size range of normal, WT synaptic vesicles*.

*Based on these measurements, a clear cutoff should be defined between synaptic vesicles and bulk endosomes (to be used in analysis figures such as*
Figure 2*)*.

This question was answered above.

*Along the same lines, the authors need to determine whether the small vesicles forming in DKO synapses exhibit the physiological characteristics of synaptic vesicles. For example, can these vesicles exocytose upon stimulation? If not, then this pathway cannot be suggested to form true synaptic vesicles. A number of strategies could be tried here, including pHluorin experiments and FM dye uptake and release*.

We addressed this question with a new experimentation (as answered above), and found that the vesicles can undergo exocytosis.